# PowerPM: Foundation Model for Power Systems

**Shihao Tu**[*]
Zhejiang University
shihao.tu@zju.edu.cn

**Yupeng Zhang**[*]
Zhejiang University
yuppzhang@zju.edu.cn

**Jing Zhang**
Renmin University of China
zhang-jing@ruc.edu.cn

**Zhendong Fu**
Zhejiang University
zhendongfu@zju.edu.cn

**Yin Zhang**
Zhejiang University
yinzh@zju.edu.cn

**Yang Yang**[†]
Zhejiang University
yangya@zju.edu.cn

## Abstract

The proliferation of abundant electricity time series (ETS) data presents numerous opportunities for various applications within power systems, including demand-side management, grid stability, and consumer behavior analysis. Deep learning models have advanced ETS modeling by effectively capturing sequence dependence. However, learning a generic representation of ETS data for various applications is challenging due to the inherently complex hierarchical structure of ETS data. Moreover, ETS data exhibits intricate temporal dependencies and is susceptible to the influence of exogenous variables. Furthermore, different instances exhibit diverse electricity consumption behavior. In this paper, we propose a foundation model PowerPM for ETS data, providing a large-scale, off-the-shelf model for power systems. PowerPM consists of a temporal encoder and a hierarchical encoder. The *temporal encoder* captures temporal dependencies within ETS data, taking into account exogenous variables. The *hierarchical encoder* models correlations between different levels of hierarchy. Furthermore, PowerPM leverages a novel self-supervised pre-training framework consisting of *masked ETS modeling* and *dual-view contrastive learning*. This framework enables PowerPM to capture temporal dependency within ETS windows and aware the discrepancy across ETS windows, providing two different perspectives to learn generic representation. Our experiments span five real-world scenario datasets, including both private and public data. Through pre-training on massive ETS data, PowerPM achieves SOTA performance on diverse downstream tasks within the private dataset. Notably, when transferred to public datasets, PowerPM retains its edge, showcasing its remarkable generalization ability across various tasks and domains. Moreover, ablation studies and few-shot experiments further substantiate the effectiveness of our model.

## 1 Introduction

The volume of Electricity Time Series (ETS) data has recently increased rapidly due to the emergence of advanced power systems known as smart grids [10]. This abundance of data has paved the way for diverse applications in power systems, including demand-side management [22], grid stability [2] and consumer behavior analysis [49], etc. Meanwhile, these applications have spawned various tasks, as shown in Fig. 1(d). These include load forecasting [27, 4], clock anomaly detection [46], electricity theft [15] and and the detection of elderly individuals living alone [45].

---

[*]These authors contributed equally to this work.
[†]Corresponding authors.

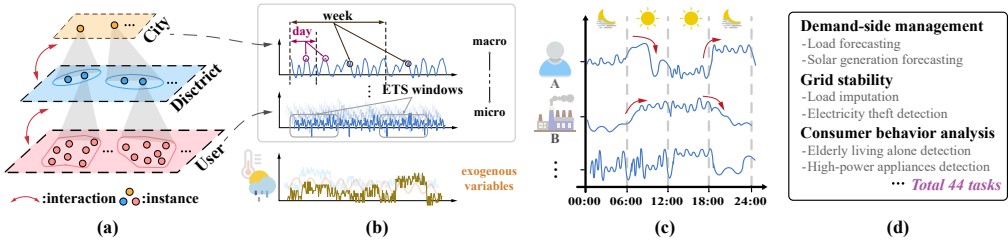

Figure 1: (a) The hierarchical structure of ETS data. (b) The temporal dependency within ETS data and the influence of exogenous variables. (c) Different electricity consumption behaviors exist across time and instances. (d) Various tasks in power systems.

As society progresses towards modernization, electricity consumption is rapidly increasing, presenting opportunities and challenges for the development and application of smart grids. On one hand, the substantial economic benefits that accompany this significant electricity usage are considerable. On the other hand, unreasonable electricity planning can have a detrimental impact on the environment[30]. Therefore, given the large volume of data and the variety of tasks, there is an urgent need to study effective ETS data modeling methods for these tasks, so as to improve economic efficiency while adhering to low-carbon principles.

Recently, numerous research studies on pre-training approaches for ETS data have emerged. These approaches adopt the "pre-training then fine-tuning" paradigm to deal with the dilemma of limited annotation data, and the pre-trained model to easily adapt to new tasks, such as PatchTST [21], TS2Vec [42], CoST [37], etc. However, these pre-training methods only utilize small-scale of data with a small number of instances (e.g. users), resulting in poor performance on downstream tasks. As the same time, many researcher begin to apply Large Language Models (LLMs) to assist time series modeling by using pre-trained LLM to encode time series [51] or incorporating additional descriptions related to the time series [17, 20]. Nevertheless, these models have limited ability in the power system scenario due to insufficient pre-training data of power systems and the lack of sufficient domain-specific knowledge. Additionally, none of these models are tailored for the scenario of power systems, so they neglect the unique characteristics of ETS data. Consequently, there remains a significant research gap in existing power systems literature regarding the modeling of ETS data using a foundation model.

In our scenario, the ETS data contains numerous instances and naturally exhibits a complex hierarchy [41, 23]. As depicted in Fig. 1(a), a city ETS can be disaggregated into district ETS accroding to the administrative divisions, which can further be disaggregated into user ETS in this district. For the complex hierarchy of ETS data, modeling ETS data entails the consideration of several challenges:

**(1) Hierarchical Dependency Modeling.** The hierarchy of ETS data facilitates information interaction across different granularities. Fine-grained ETS provides detailed insights into individual electricity usage, while coarse-grained ETS for districts and cities captures broader factors and indicates overall trends. For example, user-level data reflects user-specific behaviors and city-level data encompasses demographics and policy effects [29, 35]. Integrating these levels of granularity to provide both macro and micro perspectives is a complex task that requires sophisticated modeling.

**(2) Temporal Dependencies within ETS Window.** An ETS window refers to a piece of electricity time series over a period of time. The temporal dependencies within an ETS window refer to the correlations and dependencies between observations at different timestamps. As shown in Fig. 1(b), the city-level ETS exhibits daily and weekly dependency. Moreover, the temporal dependencies are often influenced by exogenous variables, such as weather, temperature, and seasonal effects. Integrating these factors into the model is challenging because their impact may interact with the temporal dynamics in complex ways. Accurately capturing the temporal dependencies with the impact of exogenous variables is a key challenge in modeling ETS data.

**(3) Discrepancy across ETS Windows.** The patterns observed in ETS windows can vary significantly across different instances and different timestamps. For instance, as shown in Fig. 1(c), residential electricity consumption (*User A*) reaches its peak in the mornings and evenings, used for lighting, appliances, and heating. However, electricity usage typically declines during the day because residents

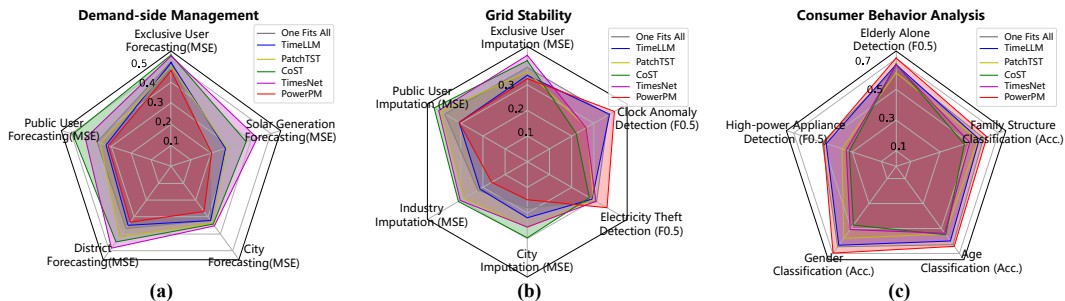

Figure 2: Performance comparison of our model and other baseline models on all downstream tasks in our scenario. Model performances are plotted on 3 radar subfigures for clarity with the same coordinate range.

are generally absent, being engaged in work or education activities outside the home. Moreover, industries (*User B*) have high power demand during specific daytime periods for machinery and production lines, with lower load requirements during nighttime and weekends. These variations in behavior highlight the challenge of achieving consistency across ETS windows in personalized modeling.

To address these challenges, we propose a foundation model for power systems named **P**ower **P**re-trained **M**odel (PowerPM), as illustrated in Figure 3. PowerPM contains about 250M parameters and is pre-trained on large-scale hierarchical ETS data with 987.42GB. Specifically, we employ the "pre-training then fine-tuning" paradigm to learn generic representations by pre-training on hierarchical ETS data and to unify various tasks by fine-tuning on downstream data. During pre-training stage, we propose a novel self-supervised pre-training framework consisting of *masked ETS modeling* and *dual-view contrastive learning*, which enables PowerPM to capture temporal dependency within ETS windows and aware the discrepancy across ETS windows, so as to provide two different perspectives to learn universal representations. PowerPM mainly consists of two modules, namely, *temporal encoder* and *hierarchical encoder*. The *temporal encoder* employs Transformer encoders to capture the temporal dependency in ETS data, and incorporates exogenous variables to make the modeling process more robust. Moreover, to model hierarchical dependency, *hierarchical encoder* utilizes R-GCN [25] to propagate information about the correlation between hierarchy. According to the message that passes through the hierarchies, the micro and macro information can effectively assist in modeling the ETS data. In summary, the main contributions of our work include:

1. We propose a foundation model for power systems named PowerPM, which is pre-trained on large-scale ETS data and provide an off-the-shelf model for power systems.

2. To the best of our knowledge, PowerPM is the first to date that considers temporal dependency and hierarchical dependency simultaneously. In addition, we present a novel self-supervised pre-training framework that combines masked ETS modeling and dual-view contrastive learning, enhancing the model's ability to learn temporal dependencies within ETS windows and aware the discrepancy across ETS windows.

3. Extensive experiments show that PowerPM generalizes well to 44 downstream tasks. Fig. 2 summarizes the results of all the downstream tasks, showing its great potential in ETS data modeling. Moreover, when transferred to the public dataset, PowerPM maintains its superiority, showcasing its remarkable generalization ability across various tasks and domains. Further analysis illustrates the effectiveness of PowerPM as well.

## 2   Methodology

**Overview.**  As shown in the middle part of Fig. 3: Firstly, the hierarchical graph $\mathcal{G}$ is constructed according to the naturally existing hierarchical relationship of ETS data. The ETS windows in $\mathcal{G}$ and its corresponding exogenous variables are denoted as $\{\boldsymbol{x}_i\}_{i=1}^N$ and $\{\boldsymbol{o}_i\}_{i=1}^N$, where $N$ is the number of instances, $\boldsymbol{x}_i \in \mathbb{R}^{T_w}$, $\boldsymbol{o}_i \in \mathbb{R}^{T_w \times K}$, and each instance ETS window spans $T_w$ time points starting at $T_a$ and ending at $T_b$. Each time point has $K$ kinds of exogenous variables. Our objective is to perform

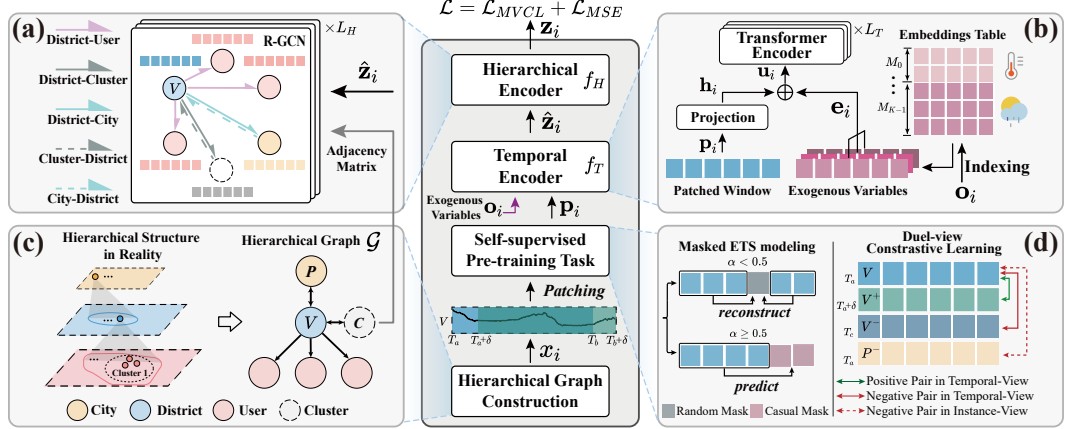

Figure 3: The pre-training framework of PowerPM. For simplicity, we take the windows of each instance in the same time range for illustration, and the window process at other times is the same.

pre-training on an encoder $f(\cdot)$ to encode each window into a latent representation $\mathbf{z}_i \in \mathbb{R}^{N \times d}$, where $d$ indicates the dimension of the latent representation. More specific, PowerPM consists of an exogenous variable enhanced temporal encoder $f_T(\cdot)$ and a hierarchical encoder $f_H(\cdot)$, with the process: $\mathbf{z}_i = f(\boldsymbol{x}_i, \boldsymbol{o}_i, \mathcal{G}) = f_H(f_T(\boldsymbol{x}_i, \boldsymbol{o}_i), \mathcal{G})$. In addition, a novel self-supervised strategy which combines masked ETS modeling and dual-view contrastive learning is used for pre-training PowerPM. Next, we will detail the techniques in both model architecture and pre-training strategy.

## 2.1 Hierarchical Graph Construction

The data of cities, districts, and users in ETS data naturally form a hierarchical relationship, based on which we can construct a hierarchical graph. However, the imbalance in the number of users and districts means there will be multitude of edges between user nodes and district nodes, which significantly increases the complexity of graph modeling. To address this, we employ a clustering strategy to create intermediary nodes, which is a common approach to implement graph sparsification [13] and a user group policy in the power systems [36, 44, 12]. As depicted in Fig. 3 (c), we use clustering method to categorize users into several clusters, the detailed process can be found in App. B.1. The cities are bidirectionally connected to districts, and these user clusters are also bidirectionally connected to districts but are unidirectionally connected to districts. By sparsifying the edges, we enhance the efficiency of graph modeling. Mathematically, we represent the hierarchy as a directed graph $\mathcal{G} = (\mathcal{V}, \mathcal{E}, \mathcal{R})$, where $\mathcal{V}$ is the set of nodes, each node corresponds to an instance, $\mathcal{E}$ is the set of directed edges, and $\mathcal{R}$ is the set of type of edges (e.g. user cluster $\rightarrow$ district, district $\rightarrow$ user, etc.).

## 2.2 Temporal Encoder with Exogenous Variables

**Patching.** In the $\mathcal{G}$, each node's feature $\boldsymbol{x}_i$ is a window of ETS data corresponding to instance $i$. Due to the semantic sparsity of time series, we patch each window $\boldsymbol{x}_i$ into $N_p$ segments, each of length $P$, resulting in $\mathbf{p}_i \in \mathbb{R}^{N_p \times P}$, where $N_p = \lceil \frac{T_w - P}{S} \rceil + 1$, and this method proved its validity in many works [21, 17, 20]. Subsequently, a linear projection is applied to each segment to obtain the window representation $\mathbf{h}_i \in \mathbb{R}^{N_p \times d}$.

**Exogenous Variables Encoding.** To efficiently interact with exogenous variables, we model these variables using learnable embeddings $\mathbf{E} \in \mathbb{R}^{(\sum_{k=0}^{K-1} M_k) \times d}$, where $K$ indicates the number of exogenous variables (e.g. weather type and temperature), $M_k$ represents the number of value types of the $k$-th exogenous variable (e.g. sunny and rainy in weather type variable). The exogenous variables $\boldsymbol{o}_i^{(k)} \in \mathbb{R}^{N_p \times P}$ corresponding to $\mathbf{p}_i$ of the $k$-th exogenous variable are used to obtain representations of the exogenous variables from $\mathbf{E}$, indexing out $\mathbf{e}_i^{(k)} \in \mathbb{R}^{N_p \times d}$, as illustrated in

Fig. 3 (b). Subsequently, we derive a representation $\mathbf{u}_i \in \mathbb{R}^{N_p \times d}$ that considers the window's exogenous variable influence: $\mathbf{u}_i = \mathbf{h}_i + \sum_{k=0}^{K-1} \mathbf{e}_i^{(k)}$.

**Temporal Encoder.** To model the complex temporal dependency and interaction with exogenous variables, we use the vanilla Transformer encoder [34] to encode $\mathbf{u}_i$, resulting in an augmented temporal representation $\hat{\mathbf{z}}_i \in \mathbb{R}^{N_p \times d}$.

## 2.3 Hierarchical Encoder

To model the complex correlation across different hierarchies, we employ Graph Neural Networks (GNNs). GNNs have recently become increasingly popular for modeling relationships within time series data, which enhances temporal representation [7, 26, 40]. In addition, considering that the correlation relationships of different edges are distinct, we adopt R-GCN [25] to integrate information across various hierarchies and instances, as depicted in Fig 3 (a). Specifically, we use R-GCN to update the representation $\hat{\mathbf{z}}$ by considering its neighboring nodes in $\mathcal{G}$, with the final node representation denoted as $\mathbf{z}_i \in \mathbb{R}^{N_p \times d}$. Moreover, we use $\mathbf{z}_i$ to perform self-supervised pre-training.

## 2.4 Self-supervised Pre-training

### 2.4.1 Masked ETS Modeling

To model temporal dependency within an ETS window, we have adopted the widely utilized masked reconstruction strategy. Nevertheless, existing random masking methods may face a significant challenge: they reconstruct the missing part based on the known surrounding part [21, 8], without considering the prediction of future parts relying solely on the past part. This approach not only diminishes the difficulty of the pre-training stage but also lacks consistency across pre-training task and forecasting task.

To address this issue, we propose a novel masking approach that combines random and casual masking, as shown in Fig. 3 (d) (*left*). Specifically, we randomly select one of the masking approaches for a given patched window $\mathbf{p}_i$, resulting in **masked** $\mathbf{p}_i$. This approach not only retains the benefits of the random masking strategy but also ensures that the model learns to predict future parts based solely on past information, thereby it can more comprehensively capture the temporal dependencies within

a window. Mathematically, this can be formulated as: **masked** $\mathbf{p}_i = \begin{cases} \text{Mask}_r(\mathbf{p}_i) & \text{if } \alpha < 0.5 \\ \text{Mask}_c(\mathbf{p}_i) & \text{otherwise} \end{cases}$,

where $\text{Mask}_r$ and $\text{Mask}_c$ denote the random and causal masking, respectively, and $\alpha \in [0, 1]$ is a uniformly distributed variable. Specifically, after the $\boldsymbol{x}_i$ is inputted into PowerPM for masked ETS modeling, we will obtain a reconstructed $\hat{\boldsymbol{x}}_i$. The corresponding reconstruction loss is: $\mathcal{L}_{MSE} = \frac{1}{N} \sum_{i=1}^{N} (\boldsymbol{x}_i - \hat{\boldsymbol{x}}_i)^2$.

### 2.4.2 Dual-view Contrastive Learning

The objective of contrastive learning is to learn representations by bringing positive pairs closer and pushing negative pairs farther apart in the latent space [5, 6]. Motivated by this, to make PowerPM aware of the discrepancy across ETS windows, we employ dual-view contrastive learning (DVCL) to discern subtle differences in electricity usage behavior.

**Positive and Negative Sample Pairs.** These pairs are determined from two views: one is *temporal view*, which is based on the time difference between the two windows. Another is the *instance view*, which depends on whether two windows belong to the same instance. For the same instance, the closer the time difference between two windows, the closer their representations are likely to be. This idea is also presented in [31, 42]. Conversely, windows from different instances or the same instance with a larger time difference are likely to have more distinct representations. Overall, we consider adjacent windows from the same instance as positive samples, while windows from different instances or non-adjacent windows from the same instance are negative samples. As depicted in Fig. 3 (d) (*right*), for the district node $\boldsymbol{V}$ in $\mathcal{G}$, the original start timestamp about this window is $T_a$. After shifting several time steps $\delta$ on, we obtain another window $V^+$ starting at $T_a + \delta$, which serves as a positive sample. Meanwhile, we select windows from other nodes in $\mathcal{G}$, such as city $\boldsymbol{P}$, starting at $T_a$, as well as windows from the same node $\boldsymbol{V}$ but starting at $T_c$, where $|T_c - T_a| \gg \delta$. These windows serve as instance and temporal negative samples, respectively, and are denoted as $P^-$ and $V^-$.

Mathematically, given an ETS window $\boldsymbol{x}_i$, we obtain a positive sample $\boldsymbol{x}_i^+$ by shifting it by $\delta$ time steps. The other samples in this batch serve as negative samples, totaling $B-1$ negative samples, where $B$ is the batch size during pre-training. The DVCL loss is: $\mathcal{L}_{DVCL} = -\sum_{i=1}^{N} \log \frac{\exp\left(\text{sim}\left(f(\boldsymbol{x}_i), f(\boldsymbol{x}_i^+)\right)/\tau\right)}{\sum_{m=1}^{B} \mathbf{I} \cdot \exp(\text{sim}(f(\boldsymbol{x}_i), f(\boldsymbol{x}_m))/\tau)}$, where $\mathbf{I}$ is the boolean vector to select the negative pairs and $\text{sim}(\cdot)$ is cosine similarity function.

## 3 Experiments

### 3.1 Experiment Setup

**Pre-training Dataset.** PowerPM is pre-trained on a mount of ETS data, a private dataset from the real scenario[1]. This pre-training dataset encompasses ETS data of cities, districts, and users, covering over 3 years records. The ETS data is collected at a frequency of one data point every 15 minutes. More details are in App. A

**Downstream Dataset.** To evaluate the performance of PowerPM, we conduct comprehensive experiments on eleven downstream private and public datasets. And seven private datasets are also collected from real scenario. These datasets have different labels for different tasks. Among them, the solar generation dataset does not have a hierarchical structure due to its particularity. Four public datasets are obtained from CSISO[2], ISONE[3], NYISO[4], and PJM[5], and they all exhibit a hierarchical structure. Further details can be found in Appendix A.

**Settings.** For the model configurations, the temporal encoder contains a 26-layer Transformer encoder with model dimension 1024, inner dimension (FFN) 2048 and 16 attention heads, and the hierarchical encoder contains 2-layer R-GCN. PowerPM contains about 250M parameters. During pre-training, the 40% segments in each input window are masked in the form of random mask and casual mask, the user cluster numbers is set to 12. See further details in App. B.1

**Baselines.** We compare with 8 state-of-the-art methods: Large Language Model (LLM) enhanced models: GPT4TS [51], Time-LLM [17], UniTime [20]; pre-train models: PatchTST [21], CoST [37], TS2Vec [42]; supervised models: DLinear [43], TimesNet [38]. More implementation details are provided in App. B.2.

**Evaluation Metrics.** For forecasting and imputation tasks, we use mean squared error (MSE): $\frac{1}{n} \sum_{i=1}^{n} (\boldsymbol{y} - \hat{\boldsymbol{y}})^2$ and mean absolute error (MAE): $\frac{1}{n} \sum_{i=1}^{n} |\boldsymbol{y} - \hat{\boldsymbol{y}}|$ as the evaluation metric. For classification tasks, we use accuracy as the metric. The metric of the anomaly detection task includes precision, recall, $F0.5$, and $F1$ scores. The $F\beta$ is a metric defined as the weighted harmonic mean of precision and recall, with the following equation: $F\beta = \frac{(1+\beta^2) \times precision \times recall}{\beta^2 \times precision + recall}$. We use $F0.5$ for anomaly detection, since precision is more important than recall in power systems scenario [15].

### 3.2 Downstream Tasks

**Demand-side Management.** Demand-side management aims to optimize and balance the power system by managing and adjusting the electricity demand of end-users. We develop tasks to predict load at different levels (such as cities and users) and tasks to forecast solar generation. With demand-side management, we can better plan and schedule power resources, improve energy efficiency, promote the development of renewable energy, and achieve sustainable energy management.

**Grid Stability.** To ensure the stability of the power grid, we have implemented a series of tasks, including electricity theft detection, load imputation, and clock anomaly detection, to address the impact of potential appliance failures within the grid and external electricity theft on the quality of power data and grid operations. Internal appliance malfunctions within the grid such as clock anomalies or the inability to record electricity usage accurately decrease the accuracy of power data, making it challenging for power dispatch and management. Additionally, external electricity theft

---

[1]Due to privacy concerns of the dataset and the company, we mask the specific information.

[2]http://www.energyonline.com/Data/

[3]https://www.iso-ne.com/isoexpress/web/reports/load-and-demand/

[4]https://www.nyiso.com/load-data

[5]https://dataminer2.pjm.com/list

can cause economic losses and pose a threat to the stable operation and reliability of the power grid, potentially causing power outages and other adverse effects.

**Consumer Behavior Analysis.** To provide users with more assistance, we have implemented tasks such as detection of elderly living alone, high-power appliance detection, gender classification, age classification, and family structure classification. Additionally, we can provide more flexible power scheduling plans for special groups, so as to optimize power dispatch. We also aim to understand the energy usage differences among different genders and age groups and provide personalized energy management recommendations and services for different users.

Table 1: Performance comparison on private dataset. The result of MAE metric refer to Tab. 6

| Tasks | | | PowerPM | PowerPM$_{freeze}$ | GPT4TS [51] | TimeLLM [17] | UniTime [20] | PatchTST [21] | CoST [37] | TS2Vec [42] | TimesNet [38] | DLinear [43] |
|---|---|---|---|---|---|---|---|---|---|---|---|---|
| | | | MSE | MSE | MSE | MSE | MSE | MSE | MSE | MSE | MSE | MSE |
| Demand-side Management | Exclusive User Forecasting | 4 | **0.3378** | 0.3557 | 0.4102 | *0.3923 | 0.4165 | 0.3929 | 0.4197 | 0.4891 | 0.4335 | 0.4228 |
| | | 96 | **0.4183** | 0.4354 | 0.4682 | 0.4832 | *0.4514 | 0.4600 | 0.5166 | 0.5453 | 0.5123 | 0.5398 |
| | | 288 | **0.4770** | 0.5026 | 0.5319 | 0.5207 | 0.5370 | *0.5173 | 0.5634 | 0.5679 | 0.5569 | 0.5818 |
| | | 672 | 0.5476 | 0.5831 | 0.5840 | *0.5789 | 0.5899 | **0.5347** | 0.6088 | 0.6013 | 0.5961 | 0.6301 |
| | | Avg. | **0.4452** | 0.4692 | 0.4986 | 0.4938 | 0.4987 | *0.4762 | 0.5271 | 0.5509 | 0.5247 | 0.5436 |
| | Public User Forecasting | 4 | **0.2353** | 0.2507 | 0.3044 | *0.2857 | 0.2967 | 0.2911 | 0.4076 | 0.3598 | 0.3583 | 0.3592 |
| | | 96 | **0.2604** | *0.3142 | 0.3456 | 0.3021 | 0.3645 | 0.3211 | 0.4395 | 0.4054 | 0.3974 | 0.4567 |
| | | 288 | **0.3226** | *0.3478 | 0.3914 | 0.3449 | 0.4050 | 0.3735 | 0.5128 | 0.5276 | 0.4359 | 0.5455 |
| | | 672 | 0.3818 | *0.4061 | 0.4470 | **0.3720** | 0.4424 | 0.4325 | 0.5565 | 0.5756 | 0.5271 | 0.5960 |
| | | Avg. | **0.3000** | *0.3297 | 0.3721 | 0.3262 | 0.3772 | 0.3546 | 0.4791 | 0.4671 | 0.4297 | 0.4894 |
| | District Forecasting | 4 | **0.2382** | 0.2736 | 0.3239 | *0.2924 | 0.3115 | 0.3489 | 0.3837 | 0.3989 | 0.4135 | 0.3701 |
| | | 96 | **0.2926** | 0.3348 | 0.3521 | *0.3434 | 0.3532 | 0.3891 | 0.4166 | 0.4507 | 0.4742 | 0.4413 |
| | | 288 | **0.3300** | *0.3760 | 0.3836 | 0.3656 | 0.3903 | 0.4458 | 0.4455 | 0.4836 | 0.4950 | 0.5186 |
| | | 672 | **0.3710** | 0.4199 | *0.4110 | 0.3940 | 0.4213 | 0.4852 | 0.5109 | 0.5402 | 0.5513 | 0.6004 |
| | | Avg. | **0.3080** | *0.3511 | 0.3677 | 0.3489 | 0.3691 | 0.4173 | 0.4392 | 0.4684 | 0.4835 | 0.4826 |
| | City Forecasting | 4 | **0.1725** | 0.2213 | 0.2754 | 0.2620 | *0.2435 | 0.2654 | 0.2757 | 0.2650 | 0.2455 | 0.3442 |
| | | 96 | **0.2272** | 0.2818 | 0.2958 | 0.2885 | 0.2910 | *0.2858 | 0.3065 | 0.2894 | 0.3030 | 0.4084 |
| | | 288 | **0.2484** | 0.3371 | 0.3311 | 0.3390 | *0.3365 | 0.3682 | 0.3540 | 0.3468 | 0.3976 | 0.4471 |
| | | 672 | **0.3211** | 0.3706 | 0.3746 | 0.3933 | *0.3727 | 0.4256 | 0.4313 | 0.4646 | 0.4622 | 0.5196 |
| | | Avg. | **0.2423** | 0.3027 | 0.3192 | 0.3207 | *0.3109 | 0.3363 | 0.3419 | 0.3415 | 0.3521 | 0.4298 |
| | Solar Generation Forecasting | 4 | **0.0993** | 0.1131 | 0.1219 | 0.1315 | 0.1561 | *0.1188 | 0.1678 | 0.2330 | 0.3379 | 0.4177 |
| | | 96 | **0.1223** | 0.1646 | 0.1894 | 0.2183 | 0.2468 | *0.1766 | 0.3822 | 0.3394 | 0.4216 | 0.4710 |
| | | 288 | **0.2337** | 0.2679 | 0.2330 | 0.2862 | 0.3366 | *0.2538 | 0.4568 | 0.3958 | 0.4570 | 0.5472 |
| | | 672 | 0.3076 | *0.3438 | **0.2893** | 0.3561 | 0.3843 | 0.3607 | 0.4984 | 0.4259 | 0.5128 | 0.5993 |
| | | Avg. | **0.1907** | *0.2224 | 0.2084 | 0.2480 | 0.2810 | 0.2275 | 0.3763 | 0.3485 | 0.4323 | 0.5088 |
| Grid Stability | Exclusive User Imputation | 0.125 | 0.2459 | 0.2832 | 0.2902 | **0.2442** | *0.2673 | 0.2820 | 0.3243 | 0.3636 | 0.3334 | 0.3702 |
| | | 0.25 | **0.2621** | *0.3136 | 0.3448 | 0.3036 | 0.3398 | 0.3318 | 0.3615 | 0.4150 | 0.3882 | 0.4139 |
| | | 0.375 | **0.3288** | 0.3573 | 0.4025 | 0.3754 | 0.4080 | *0.3725 | 0.4105 | 0.4595 | 0.4275 | 0.4634 |
| | | 0.5 | **0.3661** | 0.4125 | 0.4342 | 0.4243 | 0.4393 | *0.4190 | 0.4805 | 0.5036 | 0.5103 | 0.5365 |
| | | Avg. | **0.3007** | *0.3417 | 0.3679 | 0.3369 | 0.3636 | 0.3513 | 0.3942 | 0.4354 | 0.4149 | 0.4460 |
| | Public User Imputation | 0.125 | **0.2348** | *0.2651 | 0.2897 | 0.2614 | 0.2987 | 0.3070 | 0.3516 | 0.3223 | 0.3006 | 0.3544 |
| | | 0.25 | **0.2776** | *0.2949 | 0.3327 | 0.2837 | 0.3340 | 0.3667 | 0.4011 | 0.3888 | 0.3583 | 0.4013 |
| | | 0.375 | **0.3237** | *0.3320 | 0.4005 | 0.3044 | 0.3505 | 0.4105 | 0.4420 | 0.4316 | 0.4136 | 0.4487 |
| | | 0.5 | 0.3919 | *0.4295 | 0.4623 | **0.3776** | 0.4439 | 0.4423 | 0.4846 | 0.5028 | 0.5235 | 0.5497 |
| | | Avg. | 0.3070 | *0.3304 | 0.3713 | **0.3068** | 0.3568 | 0.3816 | 0.4198 | 0.4114 | 0.3990 | 0.4385 |
| | District Imputation | 0.125 | **0.0811** | 0.1212 | *0.1225 | 0.1364 | 0.1653 | 0.1506 | 0.1852 | 0.2222 | 0.1766 | 0.2332 |
| | | 0.25 | **0.1284** | 0.1689 | 0.2016 | *0.1710 | 0.2698 | 0.2679 | 0.2881 | 0.3042 | 0.2669 | 0.2810 |
| | | 0.375 | **0.1666** | 0.2223 | 0.2430 | *0.2381 | 0.3132 | 0.3272 | 0.3432 | 0.3524 | 0.3598 | 0.3409 |
| | | 0.5 | **0.2269** | 0.2938 | 0.3238 | *0.3068 | 0.3591 | 0.3938 | 0.4249 | 0.4227 | 0.4053 | 0.4051 |
| | | Avg. | **0.1508** | 0.2016 | 0.2227 | *0.2131 | 0.2769 | 0.2849 | 0.3104 | 0.3254 | 0.3022 | 0.3151 |
| | City Imputation | 0.125 | **0.0753** | *0.1250 | 0.1101 | 0.1465 | 0.1502 | 0.1807 | 0.2161 | 0.2476 | 0.1825 | 0.2542 |
| | | 0.25 | **0.1114** | *0.1626 | 0.1524 | 0.1912 | 0.2047 | 0.2313 | 0.2715 | 0.2885 | 0.2237 | 0.2987 |
| | | 0.375 | **0.1451** | 0.2155 | *0.2175 | 0.2409 | 0.2557 | 0.2714 | 0.3262 | 0.3313 | 0.2740 | 0.3663 |
| | | 0.5 | 0.2412 | *0.2623 | **0.2357** | 0.2965 | 0.3034 | 0.3417 | 0.3728 | 0.3935 | 0.3389 | 0.4134 |
| | | Avg. | **0.1433** | *0.1914 | 0.1789 | 0.2188 | 0.2285 | 0.2563 | 0.2967 | 0.3152 | 0.2548 | 0.3332 |
| | Electricity Theft Detection | Pre. | **0.3793** | 0.3213 | 0.2865 | 0.2537 | 0.2515 | 0.2678 | *0.3149 | 0.3076 | 0.2790 | 0.2603 |
| | | Rec. | **0.5911** | 0.5487 | 0.4444 | 0.4991 | 0.5009 | 0.4665 | *0.5281 | 0.4943 | 0.4448 | 0.4594 |
| | | F0.5 | **0.4086** | 0.3503 | 0.3084 | 0.2814 | 0.2793 | 0.2927 | *0.3426 | 0.3327 | 0.3015 | 0.2850 |
| | | F1 | **0.4621** | 0.4053 | 0.3484 | 0.3364 | 0.3349 | 0.3403 | *0.3945 | 0.3792 | 0.3429 | 0.3323 |
| | Clock Anomaly Detection | Pre. | **0.4540** | 0.3874 | 0.3108 | 0.3108 | 0.3294 | 0.2321 | 0.3620 | *0.3859 | 0.2341 | 0.1719 |
| | | Rec. | **0.7881** | 0.7391 | 0.7255 | 0.7120 | 0.6908 | 0.6290 | 0.7309 | *0.7326 | 0.5571 | 0.5432 |
| | | F0.5 | **0.4961** | 0.4281 | 0.3650 | 0.3503 | 0.3679 | 0.2656 | 0.4026 | *0.4262 | 0.2648 | 0.1991 |
| | | F1 | **0.5761** | 0.5083 | 0.4486 | 0.4327 | 0.4842 | 0.3391 | 0.4842 | *0.5055 | 0.3297 | 0.2612 |
| Consumer Behavior Analysis | High Power Appliance Detection | Pre. | **0.7427** | *0.7265 | 0.6951 | 0.6988 | **0.7430** | 0.6538 | 0.6973 | 0.6880 | 0.7027 | 0.6008 |
| | | Rec. | **0.5832** | *0.5426 | 0.4924 | 0.5024 | 0.5375 | 0.4773 | 0.5715 | 0.5116 | 0.5292 | 0.4668 |
| | | F0.5 | **0.7042** | *0.6804 | 0.6422 | 0.6481 | 0.6902 | 0.6088 | 0.6679 | 0.6436 | 0.6595 | 0.5682 |
| | | F1 | **0.6534** | 0.6212 | 0.5765 | 0.5845 | *0.6238 | 0.5518 | 0.6282 | 0.5868 | 0.6037 | 0.5254 |
| | Elderly Alone Detection | Pre. | 0.4540 | *0.4374 | **0.4677** | 0.4135 | 0.4254 | 0.3301 | 0.3826 | 0.3588 | 0.3025 | 0.2282 |
| | | Rec. | **0.7881** | 0.7587 | *0.7355 | 0.6898 | 0.7044 | 0.6448 | 0.6796 | 0.6690 | 0.6934 | 0.5704 |
| | | F0.5 | 0.4961 | *0.4779 | **0.5044** | 0.4495 | 0.4620 | 0.3658 | 0.4192 | 0.3955 | 0.3409 | 0.2593 |
| | | F1 | **0.5761** | *0.5549 | 0.5718 | 0.5171 | 0.5305 | 0.4367 | 0.4896 | 0.4671 | 0.4212 | 0.3260 |
| | Gender CLS | Acc. | **0.7571** | 0.7142 | *0.6466 | 0.6340 | 0.6328 | 0.5490 | 0.6402 | 0.5960 | 0.5079 | 0.4786 |
| | Age CLS | Acc. | **0.6830** | 0.6418 | 0.6295 | 0.6001 | 0.5774 | 0.5134 | *0.6298 | 0.5864 | 0.5379 | 0.5187 |
| | Family Structure CLS | Acc. | **0.6406** | *0.6129 | 0.5974 | 0.5687 | 0.6179 | 0.5205 | 0.6062 | 0.5463 | 0.5038 | 0.4840 |

## 3.3 Main Results

**Overview.** As a foundation model for power systems, PowerPM achieves SOTA performance on various tasks when compared to other baseline models, highlighting its ability to generalize effectively across a wide range of tasks. We derive more detailed comparisons of each task in the following paragraphs, and in all tables we mark the best results in **bold**, the second-best in underlined, and the third-best in *asterisk in each column.

**Demand-side Management.** The forecasting results for load and solar generation are presented in Tab. 1 (upper part). The results cover various forecast horizons, including 4 (1 hour), 96 (1 day), 288

(3 days), and 672 (1 week). The choice of these forecast horizons holds physical significance as it aligns with real-world scenarios. The results demonstrate that not only PowerPM achieves near SOTA performance, but also PowerPM$_{freeze}$ surpasses most baseline models. This highlights the superiority of PowerPM in modeling temporal dependencies and capturing the impact of exogenous variables through the use of a *temporal encoder* and a novel *masked ETS modeling* approach. Furthermore, PowerPM attains near SOTA performance at different hierarchical levels, particularly at the macro level (district and city), highlighting the importance of modeling the hierarchical correlation within ETS data in PowerPM. Notably, among the baselines, none of the baselines capture the hierarchical correlation of ETS data, resulting in a performance decrease in comparison to PowerPM.

**Grid Stability.** To assess the efficacy of PowerPM in grid stability application, we conduct comprehensive experiments encompassing load imputation across various masked ratios $(12.5\%, 25\%, 37.5\%, 50\%)$, anomaly detection (including electricity theft and clock anomaly detection), encompassing a total of 18 tasks. The results, detailed in Tab. 1 (middle part), illustrate PowerPM's consistent superiority over all baselines, with the PowerPM$_{freeze}$ variant also surpassing the majority of baselines. Notably, in imputation tasks, PowerPM demonstrates marked superiority over other pre-trained models (such as PatchTST and CoST), underscoring the advantages of hierarchical modeling in ETS data. Furthermore, in anomaly detection tasks, as shown in Tab. 1 (middle part), our model consistently achieves near-optimal results. While GPT4TS records the highest F0.5 score among the baseline methods, attributed to its generation of GPT-2, PowerPM further enhances the F0.5 score over GPT4TS. This improvement stems from our temporal encoder's broader receptive field and the hierarchical encoder's capacity to capture hierarchical correlations across all levels, which are both pivotal for modeling ETS data.

**Consumer Behavior Analysis.** We explore two anomaly detection tasks: elderly living alone and high-power appliance detection, and three classification tasks: gender, age, and family structure classification. The results in Tab. 1 (bottom part) demonstrate PowerPM's SOTA performance, illustrating its capacity for deep semantic insight and contextual awareness. Furthermore, PowerPM$_{freeze}$ sustains high performance, highlighting the model's innate ability to extract and generalize features.

Table 2: Performance comparison on 4 public dataset.

| Dataset | Task | | PowerPM | PowerPM$_{freeze}$ | GPT4TS [51] | TimeLLM [17] | UniTime [20] | PatchTST [21] | CoST [37] | TS2Vec [42] | TimesNet [38] | DLinear [43] |
|---|---|---|---|---|---|---|---|---|---|---|---|---|
| | | | MSE | MSE | MSE | MSE | MSE | MSE | MSE | MSE | MSE | MSE |
| CAISO | State Forecasting | 12 | 0.2968 | 0.3162 | 0.3519 | 0.3620 | 0.3187 | *0.3167 | 0.3565 | 0.4143 | 0.3604 | 0.4173 |
| | | 24 | 0.3341 | 0.3742 | 0.3857 | *0.3708 | 0.3765 | 0.3647 | 0.4151 | 0.4531 | 0.4205 | 0.4887 |
| | | 168 | 0.3767 | 0.3967 | 0.4138 | *0.4097 | 0.4211 | 0.4099 | 0.4531 | 0.5117 | 0.4754 | 0.5591 |
| | | Avg. | 0.3359 | 0.3624 | 0.3838 | 0.3808 | 0.3721 | *0.3637 | 0.4082 | 0.4597 | 0.4188 | 0.4884 |
| | Area Forecasting | 12 | 0.1877 | 0.2195 | *0.2233 | 0.2318 | 0.2528 | 0.2688 | 0.2993 | 0.3049 | 0.3401 | 0.3838 |
| | | 24 | 0.2072 | 0.2425 | *0.2478 | 0.2551 | 0.2735 | 0.3098 | 0.3320 | 0.3280 | 0.3869 | 0.4386 |
| | | 168 | 0.2645 | *0.3104 | 0.2980 | 0.3135 | 0.3344 | 0.3318 | 0.3889 | 0.3960 | 0.4259 | 0.4773 |
| | | Avg. | 0.2198 | *0.2575 | 0.2564 | 0.2668 | 0.2869 | 0.3035 | 0.3401 | 0.3430 | 0.3843 | 0.4332 |
| NYISO | State Forecasting | 12 | 0.0975 | *0.1128 | 0.1426 | 0.1241 | 0.1069 | 0.1212 | 0.2040 | 0.1978 | 0.1857 | 0.2386 |
| | | 24 | 0.1134 | 0.1421 | 0.1593 | *0.1430 | 0.1438 | 0.1984 | 0.2426 | 0.2666 | 0.2376 | 0.2932 |
| | | 168 | 0.1469 | *0.1812 | 0.1944 | 0.1830 | 0.1794 | 0.2046 | 0.3317 | 0.3164 | 0.2738 | 0.3751 |
| | | Avg. | 0.1193 | *0.1454 | 0.1654 | 0.1501 | 0.1434 | 0.1747 | 0.2594 | 0.2603 | 0.2323 | 0.3023 |
| | Area Forecasting | 12 | *0.0952 | 0.0946 | 0.1086 | 0.0854 | 0.1025 | 0.1462 | 0.1663 | 0.1593 | 0.1610 | 0.1985 |
| | | 24 | 0.1154 | 0.1567 | *0.1193 | 0.1077 | 0.1334 | 0.1573 | 0.2182 | 0.1915 | 0.2252 | 0.2444 |
| | | 168 | 0.1635 | 0.1772 | 0.1909 | *0.1690 | 0.1558 | 0.2310 | 0.2777 | 0.2524 | 0.2891 | 0.3399 |
| | | Avg. | 0.1247 | 0.1428 | 0.1396 | 0.1207 | *0.1306 | 0.1781 | 0.2207 | 0.2011 | 0.2251 | 0.2609 |
| ISONE | Region Forecasting | 12 | 0.1994 | *0.2328 | 0.2230 | 0.2352 | 0.2457 | 0.2821 | 0.3176 | 0.3559 | 0.3261 | 0.3665 |
| | | 24 | 0.2330 | *0.2833 | 0.2849 | 0.2761 | 0.2859 | 0.3277 | 0.3621 | 0.3986 | 0.3725 | 0.4185 |
| | | 168 | 0.3118 | 0.3509 | *0.3677 | 0.3847 | 0.3800 | 0.4130 | 0.4441 | 0.4522 | 0.4812 | 0.5006 |
| | | Avg. | 0.2481 | 0.2890 | *0.2918 | 0.2987 | 0.3039 | 0.3410 | 0.3746 | 0.4023 | 0.3933 | 0.4285 |
| | State Forecasting | 12 | 0.1289 | 0.1584 | 0.1756 | 0.1903 | *0.1616 | 0.2152 | 0.3207 | 0.2751 | 0.2290 | 0.3357 |
| | | 24 | 0.1648 | 0.2161 | *0.2132 | 0.2284 | 0.2044 | 0.2540 | 0.3725 | 0.3576 | 0.2784 | 0.3828 |
| | | 168 | 0.2201 | 0.2843 | *0.2713 | 0.2872 | 0.2705 | 0.3138 | 0.4171 | 0.4033 | 0.3547 | 0.4585 |
| | | Avg. | 0.1713 | *0.2196 | 0.2200 | 0.2353 | 0.2121 | 0.2610 | 0.3701 | 0.3453 | 0.2874 | 0.3924 |
| PJM | State Forecasting | 12 | 0.2516 | 0.2591 | 0.3054 | *0.2619 | 0.3119 | 0.3495 | 0.3371 | 0.3844 | 0.4056 | 0.4383 |
| | | 144 | 0.3258 | 0.3434 | 0.3834 | *0.3571 | 0.4006 | 0.4197 | 0.3937 | 0.4425 | 0.4380 | 0.4833 |
| | | 288 | 0.4094 | 0.4646 | 0.4312 | 0.4497 | 0.4505 | 0.4502 | *0.4461 | 0.4818 | 0.4933 | 0.5328 |
| | | Avg. | 0.3289 | 0.3557 | 0.3733 | *0.3562 | 0.3877 | 0.4065 | 0.3923 | 0.4363 | 0.4457 | 0.4848 |
| | city Forecasting | 12 | 0.2853 | *0.3139 | 0.3398 | 0.2765 | 0.3283 | 0.3643 | 0.4127 | 0.4107 | 0.4246 | 0.4595 |
| | | 144 | 0.3191 | *0.3421 | 0.3663 | 0.3137 | 0.3926 | 0.4225 | 0.4359 | 0.4646 | 0.4688 | 0.4829 |
| | | 288 | 0.3853 | *0.4393 | 0.4559 | 0.3904 | 0.4517 | 0.4642 | 0.4832 | 0.5132 | 0.5001 | 0.5355 |
| | | Avg. | 0.3299 | *0.3651 | 0.3873 | 0.3269 | 0.3909 | 0.4170 | 0.4439 | 0.4629 | 0.4645 | 0.4927 |

## 3.4 Model Analysis

**Generalization Ability Analysis.** To further verify the generalization ability of PowerPM on more datasets from other domains, we evaluate PowerPM on 4 public datasets mentioned above. The results in Tab. 2 demonstrate that PowerPM outperforms nearly all SOTA methods and PowerPM$_{freeze}$ surpasses most SOTA methods, highlighting the generalization superiority of PowerPM.

**Ablation Study.** To assess the effectiveness of each component in our model, we conduct several ablation experiments. Specifically, we remove the following components from our model to examine their effects on performance: the hierarchical encoder (PowerPM-H), the dual-view contrastive learning strategy (PowerPM-C), and the exogenous variables encoding module (PowerPM-E). Furthermore, we replace the masked ETS modeling module with vanilla random masking (PowerPM-M). We categorize 44 tasks into four traditional time series tasks: forecasting, missing value imputation, anomaly detection, and classification. The evaluation metrics are Mean Squared Error (MSE) for forecasting and missing value imputation, F0.5 score is for anomaly detection, and accuracy (Acc.) for classification. The performance is averaged to provide a comprehensive assessment.

The results of the ablation study are in Fig. 4 (a). The results indicate that PowerPM outperforms its vari-

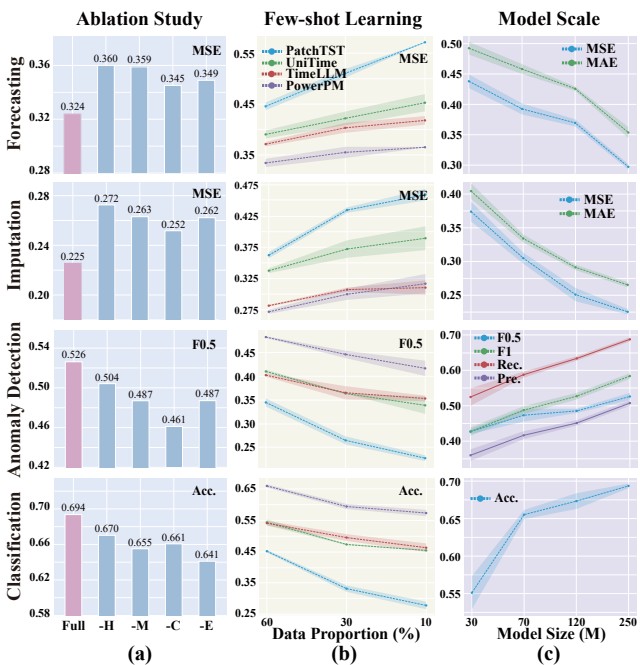

Figure 4: Model Analysis: Ablation Study, Few-shot Learning, and Model Scale Evaluation

ants, providing evidence for the contribution of each component. Among them, PowerPM-H exhibits the most substantial decrease in performance compared to the full PowerPM, emphasizing the significance of interactions between micro- and macro-levels when modeling hierarchical ETS data. The observed performance degradation of PowerPM-M, particularly in forecasting tasks, shows that causal masking can capture more complex temporal dependency. Moreover, the declined performance of PowerPM-C, particularly in anomaly detection and classification tasks, suggests that dual-view contrastive learning is effective in capturing subtle discrepancies between instances. Furthermore, PowerPM-E also presents performance degradation. This emphasizes the effectiveness of the exogenous variables encoding module in capturing the impact of exogenous factors. For detailed results of 44 tasks, please refer to App. 7.

**Few-shot Learning.** In power systems, collecting abundant ETS data for downstream tasks is a significant investment. To demonstrate the practical application value of our work, we conduct a performance comparison between PowerPM and baseline models, considering the limited availability of ETS data. Specifically, models are fine-tuned on 10%, 30% and 60% of the downstream dataset, respectively. Similar to an ablation study, we group our results by task type, which can be seen in Fig. 4 (b). The performance of PowerPM exhibits a slight decrease when there is a significant reduction in the proportion of fine-tuning data. This observation serves as evidence of the effectiveness of our novel pre-training strategy. Additionally, it highlights that the PowerPM adeptly captures temporal dependencies and hierarchical correlations present in the ETS data during pre-training, enabling easier adaptation to downstream tasks. More detailed results can be referred in App. 8.

**Model Scale Evaluation.** To explore the impact of model size on performance, we design three variants of PowerPM (about $250M$) with smaller sizes: PowerPM-Tiny (about $30M$), PowerPM-Small (about $70M$), PowerPM-Medium (about $120M$), and pre-train them on the same datasets. For the pre-training details, please refer to App. B.1. After pre-training, we evaluate these variants on all downstream tasks and present the results by task type like the ablation study. As shown in Fig. 4 (c), as the size of the model increases, we observe an overall improvement of the performance on all downstream tasks. Specifically, PowerPM outperforms the other variants in all metrics. In addition, larger models exhibit almost a decrease in standard deviation, indicating a more stable performance.

Therefore, the utilization of a larger model with higher capacity and large ETS data enables better generalization across a wide range of downstream tasks.

## 4 Related Work

**Self-supervised Pre-training Model.** Large-scale model based on self-supervised pre-training has become more significant in both industrial and academic domains due to the versatility and impressive performance. It initially developed in the fields of computer vision [14] and natural language processing [8, 11]. Self-supervised pre-training in time series is typically classified into two paradigms: contrastive learning and mask modeling. The objective of contrastive learning is to learn representation by pushing positive pairs closer and pull negative pairs away in the embedding space [16]. TS2Vec [42] proposes contextual consistency for positive pair selection. Then, CoST [37] extracts the trend and seasonal feature representations, and takes advantage of both time and frequency domain contrastive loss to encourage discriminative seasonal representation. And TF-C [47] applies time-frequency consistency for embedding time-based and frequency-based neighbors. In mask modeling, to extract the contextual semantic information, PatchTST [21] masks at the series-level.

**Supervised Learning Model.** Since the self-attention mechanism in Transformer [33] showed the great ability to seize global dependencies between input and output, recently many variants have been proposed to tackle power system tasks. LogTrans [19], Informer [48] reduce the complexity by optimizing the vanilla self-attention mechanism. Autoformer [39] leverages auto-correlation mechanism to achieve series-wise representation aggregation. FEDformer [50] incorporates frequency-domain information to enhances prediction performance while reducing complexity to linear levels. Besides, DLinear [43] questions the effectiveness of transformers as it outperforms most Transformer-based SOTAs, with a simple linear model. TimesNet [38] has treated time series as a $2D$ signal and utilized a convolution-based inception net backbone to function as a comprehensive time series model.

**Large Language Models Enhanced Model.** Recently, the advancement of Large Language Models (LLMs) has opened up new horizons in time series modeling. Many LLMs, such as llama [32], GPT-3 [11], GPT-4 [1], ChatGLM [9] have the capability to capture complex dependencies and understand varied textual data, yielding sensible reasonable generation results. Therefore, many reserachers begin to apply LLMs to assist time series modeling. Time-LLM [17] and TEXT [28] employ reprogrammed input time series with text prototype embedding and incorporate textual prompts for time series. GPT4TS [51] and UniTime [20] apply fine-tuning to selected components of LLMs to improve performance in time series analysis tasks. TEMPO [3] incorporates the decomposition of time series and retrieval-based prompt design for non-stationary time series data.

However, despite numerous methods for self-supervised and supervised time series, the research on foundation models specifically designed for power systems remains relatively sparse. And LLMs are limited in power systems scenario, lacking enough textual descriptions for domain knowledge.

## 5 Conclusion

This paper introduces the PowerPM, a foundational model designed to model ETS data within power systems. PowerPM consists of a *temporal encoder* and a *hierarchical encoder*. Furthermore, PowerPM leverages a novel self-supervised pre-training framework consisting of *masked ETS modeling* and *dual-view contrastive learning*. Our experiments involve two real-world scenario datasets, comprising private and public data. Through pre-training on massive ETS data, PowerPM achieves SOTA performance on diverse downstream tasks within the private dataset. Moreover, when transferred to the public dataset, PowerPM maintains its superiority, showcasing its remarkable generalization ability across various tasks and domains. Further analysis shows the effectiveness of a foundation model in the field of power system. Also, PowerPM is an off-the-shelf model with its code and weights. This feature greatly mitigates the challenges associated with sample and label efficiency, allowing it to be directly integrated into various power system applications.

## Acknowledgments

This work was partially supported by National Natural Science Foundation of China (No. 62322606, No. 62441605).

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

Table 3: Private dataset description

| Dataset | Instance | | Samples | Output Length | Frequency | Classes |
|---|---|---|---|---|---|---|
| Pre-training | #city
#district
#user | 11
90
1530826 | 268373267040 | - | 15 minutes | - |
| Load
forecasting | #city
#district
#user | 11
90
1563730 | 109596429408 | {4, 96, 288, 672} | 15 minutes | - |
| Load
imputation | #city
#district
#user | 11
90
1563730 | 109596429408 | 672 | 15 minutes | - |
| Solar generation
forecasting | #city
#district
#user | -
-
192 | 3458400 | {4, 96, 288, 672} | 15 minutes | - |
| Electricity theft
detection | #city
#district
#user | 11
90
44077 | 279478936 | 1 | 1day | 2 |
| Clock error
detection | #city
#district
#user | 11
90
26083 | 1070142528 | 1 | 15 minutes | 2 |
| Elderly alone
detection | #city
#district
#user | 11
90
35145 | 25762488 | 1 | 1day | 2 |
| High-power
appliance detection | #city
#district
#user | 11
90
24972 | 33402144 | 1 | 1day | 2 |
| Consumer
analysis | #city
#district
#user | 11
90
29476 | 18661860 | 1 | 1day | {2, 4, 4} |

## A    Dataset Description

We conduct experiments on 5 real-world hierarchical electricity time series datasets, one of which was collected from the real scenario. The other four are collected from CSISO [6], ISONE[7], NYISO [8], and PJM [9]. Our experiments include four typical time series analysis tasks on these datasets to evaluate the effect of our approach in both in-domain and cross-domain settings: prediction, missing value imputation, anomaly detection, and classification, which include different sampling frequencies (5 minutes, 15 minutes, 1 hour, 1 day). Moreover, it covers a variety of application scenarios in power systems (load forecasting, solar generation forecasting, electricity theft detection and consumer analysis, etc.). Tab. 3 and Tab. 4 summarize the detailed descriptions of these datasets.

### A.1    Private Dataset

Private dataset is collected from the load data in the real scenario, covering the period about 6 years. Following data preprocessing, we extract a subset of the data. In order to effectively support our research objectives, we divide the dataset into 9 distinct sub-datasets. One biggest of these sub-datasets is served as the pre-training dataset, while the remaining 7 sub-datasets are utilized as downstream datasets for downstream tasks. These downstream datasets are partitioned into train, validation, and test sets according to a $6 : 2 : 2$ ratio, ensuring that the training set contain data from the earlier time period. Further details are provided below:

---

[6] http://www.energyonline.com/Data/

[7] https://www.iso-ne.com/isoexpress/web/reports/load-and-demand/

[8] https://www.nyiso.com/load-data

[9] https://dataminer2.pjm.com/list

Table 4: Public dataset description

| Dataset | Instance | | Samples | Output Length | Frequency | Time Span |
|---|---|---|---|---|---|---|
| CAISO | #state
#area | 1
34 | 305018 | {12, 24, 168} | 1 hour | 2023-04-25∼2024-04-23 |
| ISONE | #region
#state | 1
6 | 25904 | {12, 24, 168} | 1 hour | 2023-10-01∼2024-04-01 |
| NYISO | #state
#area | 1
11 | 1396992 | {12, 24, 168} | 5 minutes | 2023-03-01∼2024-03-31 |
| PJM | #state
#city | 3
22 | 212369 | {12, 144, 288} | 5 minutes | 2024-03-28∼2024-04-26 |

**Pre-training Dataset.** The pre-training dataset is derived from a subset of the private dataset, encompassing the period about $4$ years.. It consists of unlabeled data recorded at a frequency of one data point every $15$ minutes. The dataset is structured hierarchically, including information at the user, district, and city levels.

**Load Forecasting and Missing Value Imputation Dataset.** This dataset is extracted from a portion of the private dataset about $1$ years. The dataset includes hierarchical information at the user, district, and city levels, with data points recorded every $15$ minutes. For the missing value imputation task, the dataset is structured to output $672$ data points. As for the forecasting task, there are four different prediction horizons: one hour ($4$ data points), one day ($96$ data points), three days ($288$ data points), and seven days ($672$ data points).

**Solar Generation Forecasting Dataset.** The dataset is collected from many distributed photovoltaic power stations. The dataset has not a hierarchical structure, and data points are recorded at a frequency of one point every $15$ minutes. It includes four different prediction horizons: one hour, one day, three days, and seven days.

**Electricity Theft Detection Dataset.** This dataset comprises the daily electricity consumption records (in K·Wh) of users in $1$ year. For each user, the dataset includes the daily aggregate electricity usage. Within the dataset, certain users (referred to as electricity thieves) engage in unauthorized activities involving the electricity meter in order to reduce costs.

**Clock Anomaly Dataset.** This dataset comprises millions of clock error series, each representing the time deviation, compared to the standard time, and communication delay of various watt-hour meters on a weekly basis. The dataset covers the period about $8$ months.

**Elderly Living Alone Dataset.** This dataset includes the daily electricity consumption records (in K·Wh) of village users. Additionally, employees conduct extensive on-site investigations specifically targeting these users, from which we obtain labels indicating whether each user is an elderly individual living alone or not.

**High-power Appliance Detection Dataset.** This dataset consists of the daily electricity consumption records (in K·Wh) of village users. Similar to the previous dataset, on-site investigations are conducted by same method, enabling us to collect labels indicating whether each user possesses high-power appliances.

**Consumer Analysis Dataset.** This dataset contains the daily electricity consumption records (in K·Wh) of village users. Additionally, employees conducted extensive on-site investigations targeting these users, collecting statistics related to the gender of the gender of user who lives alone, the age of the resident elderly, and family structure. The gender labels of user who lives alone are: male and female, totaling two classes; the age labels for residents are: $60 \sim 70$ years old, $70 \sim 80$ years old, $80 \sim 90$ years old, and over $90$ years old, totaling four classes; the family structure labels are: $1$ people, $2 \sim 3$ people, $4 \sim 5$ people, and more than $6$ people, totaling four classes.

### A.2 Public Datasets

Four public datasets as cross-domain datasets are selected to validate the generalization ability of our model. These four datasets are named CSISO, ISONE, NYISO, and PJM, which cover 3 types different hierarchical relationships: state-area, region-state, state-city.

**CAISO.** It is sampled from California, including 34 areas loads and an aggregated load for the state, recorded every hour from April 25, 2023, to April 23, 2024. The prediction horizons include half a day (12 points), one day (24 points), and seven days (168 points).

**ISONE.** It is sampled from New England, consisting of 6 states loads and an aggregated load for the region, recorded every hour from October 1, 2023, to April 1, 2024. The prediction horizons include half a day (12 points), one day (24 points), and seven days (168 points).

**NYISO.** It is sampled from California, containing 11 areas loads and an aggregated load for the state, recorded every 5 minutes from March 1, 2023, to March 31, 2024. The prediction horizons include one hour (12 points), half a day (144 points), and one day (288 points).

**PJM.** It is sampled from 3 states: Florida, Ohio, Washington, which includes 22 cities loads and there 3 state loads, recorded every hour from March 28, 2023, to April 26, 2024. The prediction horizons include one hour (12 points), half a day (144 points), and one day (288 points).

### A.3 Exogenous Variables

We obtained weather and temperature records for all area levels in both the private and public datasets. The weather information from the private dataset is obtained from the Weather Radar[10]. Additionally, the weather information from the public datasets is obtained from the NSF NCAR Research Data Archive[11]. Both sources cover the same timespan as mentioned above, respectively. These records include the maximum and minimum temperatures (in °C for private dataset and °F for public datasets) for each hour in each city.

## B  PowerPM and Baseline Implementation Details

### B.1  PowerPM Implementation

The pre-training stage of the experiment is implemented in PyTorch [24] and conducted on a Linux system with 2 CPUs (AMD EPYC 9654 96-Core Processor) and 8 GPUs (NVIDIA Tesla $A800\ 80G$) for about 8 days. And the downstream task experiment is repeated five times. We select 512 samples as a batch, and every batch contains about $174k$ patches, which we set patch len to 48, stride to 24. To speed up the model training, we stop the gradient update of the background nodes in the hierarchical graph. We optimize with Adam [18], updating the model parameters every 4 steps, and the model trains for $1310k$ updates in total. A reduce learning rate on plateau scheduler is utilized to adjust learning rate during pre-training. Specifically, we set the basic learning rate as $1e-6$ and the maximum learning rate as $2e-5$, and the learning rate updates for every $10k$ updates. In addition, we trained three additional variants of PowerPM with different parameter counts to meet the needs of different users or situations. Detailed model hyperparameters can be found in Tab. 5.

**Full Fine-tuning.** In the F-FT (Full Fine-tuning) setup, for different tasks, we introduce different head $H$ on the top of pre-trained encoder $f(.)$, where both the parameters of the encoder $f(.)$ and the head $H$ are trainable. For forecasting and imputation tasks, we use a prediction $H_l$ head to map prediction points or reconstruction points from $\mathbf{z}_i$. In this setup, we fine-tune both the head $H$ and the encoder $f(.)$. We utilize 100%, 60%, 30% and 10% training data for fine-tuning. we utilize a one-layer fully connected network to implement prediction $H_l$ and logistic regression from the Sklearn library to implement the classifier $H_c$. The learning rates are specifically set to $4e-4$ and $3e-5$ for public and private datasets.

**Partial Fine-tuning.** In the P-FT (Partial Fine-tuning) setup, for different tasks, we also introduce different head $H$ on the top of pre-trained encoder $f(.)$. For forecasting and imputation tasks, we use a prediction $H_l$ head to map prediction points or reconstruction points from $\mathbf{z}_i$. And for anomaly

---

[10]`http://en.weather.com.cn/`
[11]`https://rda.ucar.edu/`

detection and classfication tasks, a classifier $H_c$ on top of the pre-trained encoder $f(.)$. During the whole finetune process, we keep the parameters of $f(.)$ fixed. Only the head is fine-tuned in this setup. we utilize a one-layer fully connected network to implement prediction $H_l$ and logistic regression from the Sklearn library to implement the classifier $H_c$. The learning rates are specifically set to $4e-4$ and $3e-5$ for public and private datasets.

## B.2 Baselines Implementation

We compare with 8 state-of-the-art methods: including Large Language Model (LLM) enhanced models: GPT4TS [51], Time-LLM [17], UniTime [20]; pre-train models: PatchTST [21], CoST [37], TS2Vec [42]; supervised models: DLinear [43], TimesNet [38]. To make a fair and comprehensive comparison, we reproduce all models with official implementation, and use different output head for different downstream tasks. Due to the large scale of the ETS dataset, we increase the number of training epoch and reduce the learning rate in order to make the parameters of the model fully learned.

**GPT4TS** [51] combines the LLM with Transformer, which use frozen pre-trained GPT-2 for general time series analysis. To implement GPT4TS, we utilized their open-source code, available at https://github.com/DAMO-DI-ML/NeurIPS2023-One-Fits-All. We use the 6 layers of GPT-2, which is proved to have the optimal performance in original paper and the total size of GPT4TS is about 105.15M, and the trainable parameters are $24.04M$ (GPT-2 is frozen). We set the number of train epochs to $50$, the learning rate to $0.0005$, and the batch size to $256$.

**Time-LLM** [17] frezees the LLM as the backbone, and align time series to text with patch reprogramming. It also designs Prompt-as-Prefix including dataset context, task instruction and input statistics to enrich the input context to direct the transformation of reprogrammed input. We utilized their open-source code, available at https://github.com/KimMeen/Time-LLM to implement Time-LLM. We set the llama-7b with 32 layers as the backbone, which is the most effective recorded in [17] and the total size of Time-LLM is about$7.28B$, and the trainable parameters are $58.55M$ (llama-7b is frozen). To align the dataset context input to our datasets, we constuct different natural language prompt summarized in App. A for private and public datasets, and we set the number of train epochs to $50$, the learning rate to $0.005$, and the batch size to $256$.

**UniTime** [20] leverages LLM to handle time series forecasting across time series domains, which exhibit significant differences in temporal patterns and distribution. The same as dataset context in Time-LLM, UniTime also designs human-crafted instructions to furnish the model with explicit domain identification information. To implement UniTime, we utilized their open-source code, available at https://github.com/liuxu77/UniTime. We implement the backbone LLM with GPT2-small like original paper, and the total size of UniTime is about $108.54M$ without freeze any parameters. We use the same natural language prompt in Time-LLM as the human-crafted instructions for different datasets, and we set the number of train epochs to $50$, the learning rate to $0.0005$, the weight decay to $0.0001$, and the batch size to $256$.

**TS2Vec** [42] performs contextual consistency using overlapping subseries and a hierarchical loss function to capture data consistency at the observation and sample levels. We utilize the open-source code available at https://github.com/zhihanyue/ts2vec. Specifically, we set the number of epochs for pre-training to $100$, the learning rate to $0.0005$, and the batch size to $512$. Due to the large scale and complex semantics of the pre-trained ETS data, we adjust the representation dimension to $640$, matching the ETS data characteristics. We adopt the default settings provided by the TS2Vec implementation for other settings during pre-training.

**CoST** [37] comprises both time domain and frequency domain contrastive losses to learn discriminative trend and seasonal representations. We utilize the open-source code available at https://github.com/salesforce/CoST to implement CoST. Specifically, we set the number of epochs for pre-training to $100$, the learning rate to $0.0005$, representation dimension to $640$, and the batch size to $256$. We adopt the default settings provided by the CoST implementation for other settings during pre-training.

**PatchTST** [21] changes the input sequence as a series of patch windows, focus the subseries-level attention to capture local semantic information while minimizing memory consumption. We utilize the open-source code available at https://github.com/yuqinie98/PatchTST. For hyperparameters of PatchTST, We set the patch len to $32$ and stride to $16$, the number of epochs for pre-training to $100$,

the learning rate to $0.0005$, and the batch size to $512$. We adopt the default settings provided by the PatchTST implementation for other settings during pre-training.

**TimeNet** [38] is a CNN based time series model which extends the analysis of temporal variations into the $2D$ space. It designs TimesBlock with an inception block to extract complex temporal patterns, leading to multiple time series tasks. To implement TimesNet, we utilized their open-source code, available at https://github.com/thuml/Time-Series-Library. Specifically, we set the number of epochs for training to $50$, the learning rate to $0.0005$, and the batch size to $128$. We adopt the default settings provided by the TimesNet implementation for other settings for forecasting, imputation classfication anomaly detection .

**DLinear** [43] decomposes the time series into a trend sequence and a seasonal sequence, then model these two sequences using two simple MLPs. To implement DLinear, we utilized their open-source code, available at https://github.com/cure-lab/LTSF-Linear. Specifically, we set the number of epochs for training to $50$, the learning rate to $0.0005$, and the batch size to $512$. We adopt the default settings provided by the DLinear implementation for other settings.

### B.3 Cluster Method

We use K-means algorithm to cluster users. Firstly, we get filter out user ETS by labels, and normalize the time series data, represented as an $N \times M$ matrix, to ensure that differences in scale do not affect the clustering results. Next, we use DTW as the distance metric to cope with time shifts and different rate variations in ETS data and randomly initialize a cluster centers. By calculating the distance from each time series to each cluster center, it is assigned to the nearest cluster center, and the cluster center is recalculated according to the assignment result,and the process is iterated until the cluster center is stable. We experimented 10 times with different $K$, and used elbow method to select the optimal number of clusters, and finally determined 12.

## C  Full Results

Due to the limited length of the text, we summarize all the experiments in the main text into two parts: the main experiment and the analytical experiment. We categorize and index them in Table 6, 7, 8.

## D  Limitations

PowerPM is designed for electricity time series modeling, containing about 250M parameters. As a foundation model, although we have provided relatively comprehensive results to verify the model's effectiveness, the model still exsits limitations. In fact, there are various kinds of ETS in the power systems, which contain not only the electricity consumption data generated by human activities, but also the sequence generated by system operation and sensor detection. In this paper, PowerPM only pre-train on load data. In the future, by increasing model parameters and improving model architecture, we will use more kinds of ETS data for training, so that it can capture more complicated ETS semantic information, understand more complex power system operation rules, and provide more complete help for power systems.

## E  Social Impacts

This paper presents PowerPM as a foundation model for power systems and has been deployed in the real scenario. It focus on demand-side management, grid stability and consumer behavior analysis, providing the possibility to understand and analyze electricity time series. There is no potential ethical risk or negative social impact.

Table 5: The model hyperparameters of PowerPM with different model size.

| Parameter | PowerPM | PowerPM-Medium | PowerPM-Small | PowerPM-Tiny |
|---|---|---|---|---|
| Model Scale | $256.0M$ | $120.1M$ | $68.6M$ | $35.5M$ |
| Temporal Encoder | 26 | 18 | 12 | 4 |
| Model Dimention | 1024 | 768 | 768 | 768 |
| Inner Dimension | 2048 | 2048 | 1024 | 768 |
| Hierarchical Encoder Layer | 2 | 2 | 2 | 2 |
| Heads | 16 | 16 | 16 | 16 |
| Mask Ratio | 0.4 | 0.4 | 0.4 | 0.4 |
| Time Shift $\delta$ | 96 | 96 | 96 | 96 |
| Number of Clusters $K$ | 12 | 12 | 12 | 12 |
| Batch Size | 512 | 256 | 256 | 128 |
| Learning Rate | 1e-6 | 1e-6 | 2e-6 | 2e-6 |
| Optimizer | Adam | Adam | Adam | Adam |
| Scheduler | Plateau | Plateau | Plateau | Plateau |

Table 6: Additional performance comparison on private dataset in terms of MAE metric. Forecasting tasks involve varying forecasting lengths of $\{4, 96, 288, 672\}$ time points and imputation tasks involve varying mask ratio $\{0.125, 0.25, 0.375, 0.5\}$. The length of the input window is 672.

| Tasks | | PowerPM | PowerPM$_{freeze}$ | GPT4TS [51] | TimeLLM [17] | UniTime [20] | PatchTST [21] | CoST [37] | TS2Vec [42] | TimesNet [38] | DLinear [43] |
|---|---|---|---|---|---|---|---|---|---|---|---|
| | | MAE | MAE | MAE | MAE | MAE | MAE | MAE | MAE | MAE | MAE |
| Exclusive User Forecasting | 4 | **0.3638** | 0.3762 | 0.4246 | *0.4043* | 0.4166 | 0.4286 | 0.4412 | 0.4880 | 0.4512 | 0.4640 |
| | 96 | **0.4496** | 0.4717 | *0.4582* | 0.4732 | 0.4533 | 0.4657 | 0.5357 | 0.5157 | 0.4963 | 0.5354 |
| | 288 | **0.4653** | 0.4998 | *0.4891* | 0.5012 | 0.5033 | 0.4850 | 0.5875 | 0.5651 | 0.5771 | 0.5955 |
| | 672 | 0.5222 | 0.5560 | *0.5281* | 0.5557 | 0.5330 | **0.5118** | 0.6257 | 0.6132 | 0.5362 | 0.6101 |
| | Avg. | **0.4502** | 0.4759 | *0.4750* | 0.4836 | 0.4765 | 0.4728 | 0.5475 | 0.5455 | 0.5152 | 0.5512 |
| Public User Forecasting | 4 | **0.3351** | 0.3763 | 0.4099 | *0.3848* | 0.3894 | 0.4216 | 0.4622 | 0.4307 | 0.4016 | 0.4210 |
| | 96 | **0.3590** | *0.4227* | 0.4563 | 0.4128 | 0.4326 | 0.4362 | 0.5136 | 0.4574 | 0.4315 | 0.5310 |
| | 288 | *0.4575* | 0.4957 | 0.4992 | 0.4344 | 0.4859 | 0.4511 | 0.5546 | 0.5394 | 0.4924 | 0.5915 |
| | 672 | *0.4941* | 0.5327 | 0.5362 | 0.4807 | 0.5510 | 0.4613 | 0.6125 | 0.5831 | 0.5558 | 0.6537 |
| | Avg. | **0.4114** | 0.4569 | 0.4754 | 0.4282 | 0.4647 | *0.4425* | 0.5357 | 0.5027 | 0.4703 | 0.5493 |
| District Forecasting | 4 | **0.3690** | 0.3988 | 0.4120 | *0.3938* | 0.4216 | 0.4515 | 0.4525 | 0.4690 | 0.3914 | 0.4298 |
| | 96 | **0.3719** | 0.4222 | 0.4457 | 0.4406 | *0.4343* | 0.4780 | 0.5190 | 0.5110 | 0.4614 | 0.5243 |
| | 288 | **0.4174** | 0.4733 | 0.4777 | 0.4610 | 0.4605 | 0.5288 | 0.5565 | 0.5544 | 0.5076 | 0.6161 |
| | 672 | **0.4541** | 0.4552 | 0.5138 | 0.4960 | *0.4871* | 0.5625 | 0.5916 | 0.5786 | 0.5470 | 0.6407 |
| | Avg. | **0.4031** | 0.4374 | 0.4623 | *0.4479* | 0.4509 | 0.5052 | 0.5299 | 0.5283 | 0.5283 | 0.5527 |
| City Forecasting | 4 | **0.1639** | 0.2092 | 0.2333 | *0.1850* | 0.2465 | 0.2643 | 0.3482 | 0.2962 | 0.2752 | 0.3826 |
| | 96 | **0.2131** | 0.2464 | 0.2704 | *0.2578* | 0.2654 | 0.3020 | 0.3579 | 0.3191 | 0.2911 | 0.4213 |
| | 288 | **0.2471** | 0.3099 | 0.3339 | 0.3364 | 0.3494 | 0.3514 | 0.3974 | 0.3594 | *0.3306* | 0.5142 |
| | 672 | **0.2891** | *0.3645* | 0.3885 | 0.3775 | 0.4001 | 0.3826 | 0.4202 | 0.3902 | 0.3470 | 0.5554 |
| | Avg. | **0.2283** | 0.2825 | 0.3065 | *0.2892* | 0.3154 | 0.3251 | 0.3809 | 0.3412 | 0.3110 | 0.4684 |
| Solar Generation Forecasting | 4 | 0.1541 | *0.1823* | **0.1532** | 0.2212 | 0.2296 | 0.2299 | 0.2296 | 0.2712 | 0.3913 | 0.4393 |
| | 96 | 0.2602 | *0.2714* | **0.2447** | 0.2816 | 0.2811 | 0.2925 | 0.3141 | 0.3376 | 0.4102 | 0.4727 |
| | 288 | **0.3126** | 0.3970 | 0.3384 | *0.3424* | 0.3527 | 0.3588 | 0.3853 | 0.3732 | 0.4457 | 0.5228 |
| | 672 | **0.3765** | 0.4205 | *0.3892* | 0.4058 | 0.3827 | 0.3919 | 0.4646 | 0.4418 | 0.4869 | 0.5531 |
| | Avg. | **0.2759** | 0.3178 | 0.2813 | 0.3128 | 0.3115 | 0.3183 | 0.3484 | 0.3560 | 0.4335 | 0.4970 |
| Exclusive User Imputation | 0.125 | 0.2654 | 0.3164 | 0.3101 | **0.2565** | *0.2746* | 0.3041 | 0.3419 | 0.3549 | 0.3477 | 0.3792 |
| | 0.25 | **0.2849** | 0.3039 | 0.3543 | *0.3388* | 0.3638 | 0.3597 | 0.4016 | 0.4278 | 0.3935 | 0.4268 |
| | 0.375 | **0.3017** | 0.3844 | 0.3944 | *0.3913* | 0.4313 | 0.4195 | 0.4639 | 0.4787 | 0.4239 | 0.4908 |
| | 0.5 | **0.3528** | 0.4494 | 0.4617 | 0.4587 | *0.4517* | 0.4521 | 0.5246 | 0.5449 | 0.4746 | 0.5229 |
| | Avg. | **0.3012** | 0.3635 | 0.3801 | *0.3613* | 0.3804 | 0.3839 | 0.4330 | 0.4516 | 0.4099 | 0.4549 |
| Public User Imputation | 0.125 | **0.2014** | 0.2329 | 0.2552 | *0.2469* | 0.2976 | 0.3292 | 0.4256 | 0.3648 | 0.3616 | 0.3986 |
| | 0.25 | **0.2536** | 0.2959 | 0.3236 | 0.2758 | 0.3319 | 0.3936 | 0.4650 | 0.4178 | 0.4328 | 0.4679 |
| | 0.375 | **0.2592** | 0.3613 | *0.3578* | 0.3167 | 0.3839 | 0.4578 | 0.5157 | 0.4693 | 0.5119 | 0.5447 |
| | 0.5 | **0.3618** | 0.4122 | *0.4049* | 0.3351 | 0.4275 | 0.5089 | 0.5451 | 0.5148 | 0.5387 | 0.6106 |
| | Avg. | **0.2690** | 0.3256 | 0.3354 | *0.2936* | 0.3602 | 0.4224 | 0.4879 | 0.4417 | 0.4613 | 0.5055 |
| District Imputation | 0.125 | **0.1021** | 0.1427 | *0.1624* | 0.1799 | 0.1900 | 0.1992 | 0.2469 | 0.2604 | 0.2456 | 0.2653 |
| | 0.25 | **0.1543** | 0.1782 | 0.2268 | *0.2234* | 0.2694 | 0.2976 | 0.3559 | 0.3443 | 0.3115 | 0.3406 |
| | 0.375 | **0.1904** | 0.2178 | 0.2566 | 0.2755 | 0.2983 | 0.3359 | 0.3705 | 0.3947 | 0.3580 | 0.4318 |
| | 0.5 | **0.2352** | 0.2562 | *0.3162* | 0.3576 | 0.3479 | 0.3882 | 0.4546 | 0.4451 | 0.4201 | 0.4893 |
| | Avg. | **0.1705** | 0.1987 | *0.2405* | 0.2591 | 0.2764 | 0.3052 | 0.3570 | 0.3611 | 0.3338 | 0.3818 |
| City Imputation | 0.125 | **0.0876** | *0.1439* | 0.1531 | 0.1350 | 0.1490 | 0.1901 | 0.2330 | 0.2521 | 0.2004 | 0.2715 |
| | 0.25 | **0.1294** | *0.1873* | 0.1832 | 0.2141 | 0.2240 | 0.2548 | 0.2986 | 0.2933 | 0.2753 | 0.3503 |
| | 0.375 | **0.1735** | 0.2285 | 0.2024 | 0.2524 | 0.2593 | 0.3032 | 0.3516 | 0.3438 | 0.3048 | 0.3773 |
| | 0.5 | 0.2533 | *0.3009* | **0.2437** | 0.3027 | 0.3324 | 0.3866 | 0.4350 | 0.4234 | 0.3605 | 0.4102 |
| | Avg. | **0.1610** | 0.2151 | 0.1956 | 0.2260 | 0.2412 | 0.2837 | 0.3296 | 0.3282 | 0.2853 | 0.3523 |

Table 7: Detailed performance of ablation study. Forecasting tasks involve varying forecasting lengths of $\{4, 96, 288, 672\}$ time points, imputation tasks involve varying mask ratio $\{0.125, 0.25, 0.375, 0.5\}$. The length of the input window is 672.

| | Tasks | | PowerPM | | PowerPM-H | | PowerPM-M | | PowerPM-C | | PowerPM-E | |
|---|---|---|---|---|---|---|---|---|---|---|---|---|
| | | | MSE | MAE | MSE | MAE | MSE | MAE | MSE | MAE | MSE | MAE |
| Demand-side Management | Exclusive User Forecasting | 4 | 0.3378 | 0.3638 | 0.3505 | 0.3808 | 0.3777 | 0.3859 | 0.3672 | 0.3776 | *0.3531 | *0.3788 |
| | | 96 | 0.4183 | 0.4496 | 0.4389 | *0.4642 | *0.4343 | 0.4770 | 0.4253 | 0.4546 | 0.4496 | 0.4650 |
| | | 288 | 0.4770 | 0.4653 | 0.5061 | *0.4879 | 0.4957 | 0.4906 | *0.4894 | 0.4885 | 0.4853 | 0.4718 |
| | | 672 | 0.5476 | 0.5222 | *0.5765 | 0.5494 | 0.5772 | 0.5502 | 0.5957 | 0.5362 | 0.5668 | *0.5371 |
| | | Avg. | 0.4452 | 0.4502 | *0.4680 | 0.4706 | 0.4712 | 0.4759 | 0.4694 | *0.4642 | 0.4637 | 0.4632 |
| | Public User Forecasting | 4 | 0.2353 | 0.2951 | 0.2428 | 0.3041 | 0.2793 | *0.3024 | 0.2519 | 0.3239 | *0.2448 | 0.2977 |
| | | 96 | 0.2604 | 0.3190 | 0.3126 | 0.3293 | 0.3029 | 0.3473 | *0.2973 | 0.3339 | 0.2966 | *0.3325 |
| | | 288 | 0.3226 | 0.3875 | *0.3455 | 0.4103 | 0.3480 | *0.4047 | 0.3460 | 0.3938 | 0.3334 | 0.4096 |
| | | 672 | 0.3818 | 0.4241 | 0.4330 | 0.4683 | *0.4003 | 0.4595 | 0.3946 | *0.4431 | 0.4031 | 0.4349 |
| | | Avg. | 0.3000 | 0.3564 | 0.3335 | 0.3780 | 0.3326 | 0.3785 | *0.3225 | *0.3737 | 0.3195 | 0.3687 |
| | District Forecasting | 4 | 0.2382 | 0.3090 | *0.2643 | 0.3394 | 0.2739 | *0.3222 | 0.2418 | 0.3165 | 0.2714 | 0.3232 |
| | | 96 | 0.2926 | 0.3419 | 0.3454 | 0.3913 | *0.3371 | 0.3654 | 0.3278 | 0.3699 | 0.3385 | 0.3796 |
| | | 288 | 0.3300 | 0.3874 | 0.3767 | 0.4338 | 0.3896 | 0.4015 | 0.3417 | 0.4188 | *0.3659 | 0.4190 |
| | | 672 | 0.3710 | 0.4241 | 0.4105 | 0.4757 | *0.3924 | 0.4682 | 0.3809 | 0.4485 | 0.4038 | *0.4583 |
| | | Avg. | 0.3080 | 0.3656 | 0.3492 | 0.4100 | 0.3483 | *0.3893 | 0.3231 | 0.3884 | 0.3449 | 0.3950 |
| | City Forecasting | 4 | 0.1725 | 0.1639 | *0.2054 | 0.1710 | 0.2340 | 0.1934 | 0.2123 | *0.1770 | 0.1941 | 0.1812 |
| | | 96 | 0.2272 | 0.2131 | 0.2669 | 0.2570 | *0.2462 | 0.2313 | 0.2336 | *0.2403 | 0.2478 | 0.2415 |
| | | 288 | 0.2484 | 0.2471 | 0.3187 | 0.3114 | 0.3119 | *0.2950 | 0.2670 | 0.2929 | 0.2713 | 0.3054 |
| | | 672 | 0.3211 | 0.3191 | 0.3646 | 0.3820 | 0.3415 | *0.3498 | *0.3486 | 0.3426 | 0.3563 | 0.3622 |
| | | Avg. | 0.2423 | 0.2358 | 0.2889 | 0.2804 | 0.2834 | *0.2674 | 0.2654 | 0.2632 | 0.2674 | 0.2726 |
| | Solar Generation Forecasting | 4 | 0.0993 | 0.1541 | - | - | *0.1115 | 0.1827 | 0.1117 | 0.1691 | 0.1109 | *0.1732 |
| | | 96 | 0.1223 | 0.2002 | - | - | *0.1603 | 0.2270 | 0.1412 | 0.2097 | 0.1694 | 0.2310 |
| | | 288 | 0.2337 | 0.2526 | - | - | *0.2637 | 0.2859 | 0.2548 | *0.3113 | 0.2713 | 0.3138 |
| | | 672 | 0.3076 | 0.3165 | - | - | 0.3616 | 0.3332 | 0.3213 | *0.3373 | *0.3562 | 0.3686 |
| | | Avg. | 0.1907 | 0.2309 | - | - | *0.2243 | 0.2572 | 0.2073 | 0.2569 | 0.2270 | 0.2717 |
| Grid Stability | Exclusive User Imputation | 0.125 | 0.2459 | 0.2654 | 0.2665 | 0.2999 | 0.2738 | *0.2845 | *0.2633 | 0.2717 | 0.2508 | 0.2865 |
| | | 0.25 | 0.2621 | 0.2849 | 0.3160 | 0.3165 | 0.3055 | 0.3210 | *0.3025 | 0.3117 | 0.2957 | *0.3146 |
| | | 0.375 | 0.3288 | 0.3017 | 0.3586 | 0.3555 | 0.3729 | 0.3892 | *0.3594 | 0.3359 | 0.3783 | *0.3434 |
| | | 0.5 | 0.3661 | 0.3528 | 0.4426 | 0.4095 | 0.4141 | 0.4185 | 0.4421 | *0.3840 | *0.4209 | 0.3723 |
| | | Avg. | 0.3007 | 0.3012 | 0.3459 | 0.3454 | *0.3416 | 0.3533 | 0.3418 | 0.3258 | 0.3364 | *0.3292 |
| | Public User Imputation | 0.125 | 0.2348 | 0.1514 | 0.2633 | 0.1762 | 0.2495 | *0.1777 | *0.2484 | 0.1819 | 0.2457 | 0.1841 |
| | | 0.25 | 0.2776 | 0.2036 | 0.3197 | 0.2179 | 0.2884 | 0.2101 | 0.2793 | 0.2171 | *0.2847 | *0.2168 |
| | | 0.375 | 0.3237 | 0.2392 | 0.3621 | 0.3003 | 0.3541 | 0.2943 | 0.3367 | 0.2652 | *0.3471 | *0.2716 |
| | | 0.5 | 0.3919 | 0.3418 | 0.4485 | 0.3866 | *0.4201 | 0.3734 | 0.3983 | 0.3556 | 0.4288 | 0.3566 |
| | | Avg. | 0.3070 | 0.2340 | 0.3484 | 0.2703 | 0.3280 | 0.2639 | 0.3156 | 0.2549 | *0.3265 | *0.2573 |
| | District Imputation | 0.125 | 0.0811 | 0.1021 | 0.1268 | 0.1508 | 0.1185 | 0.1496 | *0.1074 | *0.1140 | 0.1058 | 0.1073 |
| | | 0.25 | 0.1284 | 0.1543 | *0.1524 | 0.2007 | 0.1505 | 0.1843 | 0.1536 | 0.1576 | 0.1629 | *0.1676 |
| | | 0.375 | 0.1666 | 0.1904 | 0.2188 | 0.2417 | 0.2147 | *0.2330 | 0.1878 | 0.2115 | 0.2033 | 0.2556 |
| | | 0.5 | 0.2269 | 0.2452 | 0.2753 | 0.3085 | *0.2771 | 0.2905 | 0.2864 | *0.3048 | 0.3028 | 0.3155 |
| | | Avg. | 0.1508 | 0.1730 | 0.1933 | 0.2254 | *0.1902 | 0.2144 | 0.1838 | 0.1970 | 0.1937 | *0.2115 |
| | City Imputation | 0.125 | 0.0753 | 0.0876 | 0.1222 | 0.1407 | 0.1078 | 0.1208 | 0.0819 | *0.1068 | *0.0993 | 0.1009 |
| | | 0.25 | 0.1114 | 0.1294 | 0.1688 | 0.1832 | 0.1491 | 0.1549 | 0.1210 | 0.1562 | *0.1472 | 0.1651 |
| | | 0.375 | 0.1451 | 0.1735 | *0.2108 | 0.2335 | 0.2362 | *0.2136 | 0.1886 | 0.1962 | 0.2253 | 0.2140 |
| | | 0.5 | 0.2412 | 0.2533 | 0.3055 | 0.2943 | *0.2742 | *0.2715 | 0.2689 | 0.2666 | 0.2957 | 0.2844 |
| | | Avg. | 0.1433 | 0.1610 | 0.2018 | 0.2129 | *0.1918 | *0.1902 | 0.1651 | 0.1815 | 0.1919 | 0.1911 |
| | Electricity Theft Detection | Pre. | 0.3793 | | 0.3612 | | *0.3457 | | 0.3068 | | 0.3141 | |
| | | Rec. | 0.5911 | | 0.5597 | | 0.5175 | | *0.5288 | | 0.5204 | |
| | | F0.5 | 0.4086 | | 0.3888 | | *0.3703 | | 0.3349 | | 0.3412 | |
| | | F1 | 0.4621 | | 0.4391 | | *0.4145 | | 0.3883 | | 0.3918 | |
| | Clock Anomaly Detection | Pre. | 0.4540 | | 0.4437 | | *0.4462 | | 0.4178 | | 0.4469 | |
| | | Rec. | 0.7881 | | 0.7574 | | *0.7446 | | 0.7184 | | 0.7358 | |
| | | F0.5 | 0.4961 | | 0.4838 | | 0.4850 | | 0.4559 | | *0.4849 | |
| | | F1 | 0.5761 | | 0.5596 | | *0.5580 | | 0.5283 | | 0.5560 | |
| Consumer Behavior Analysis | High Power Appliance Detection | Pre. | 0.7427 | | 0.7364 | | *0.7130 | | 0.6915 | | 0.7040 | |
| | | Rec. | 0.5832 | | *0.5619 | | 0.5610 | | 0.5452 | | 0.5648 | |
| | | F0.5 | 0.7042 | | 0.6934 | | *0.6763 | | 0.6563 | | 0.6709 | |
| | | F1 | 0.6534 | | 0.6374 | | *0.6279 | | 0.6097 | | 0.6267 | |
| | Elderly Alone Detection | Pre. | 0.4540 | | *0.4097 | | 0.3737 | | 0.3588 | | 0.4121 | |
| | | Rec. | 0.7881 | | *0.7551 | | 0.7654 | | 0.6956 | | 0.7293 | |
| | | F0.5 | 0.4961 | | *0.4509 | | 0.4163 | | 0.3972 | | 0.4514 | |
| | | F1 | 0.5761 | | 0.5311 | | 0.5022 | | 0.4734 | | *0.5266 | |
| | Gender CLS | Acc. | 0.7571 | | *0.7169 | | 0.6946 | | 0.7233 | | 0.6854 | |
| | Age CLS | Acc. | 0.6830 | | 0.6671 | | 0.6515 | | 0.6470 | | *0.6562 | |
| | Family Structure CLS | Acc. | 0.6406 | | 0.6265 | | *0.6191 | | 0.6114 | | 0.5815 | |

Table 8: Complete results of few-shot learning performance comparison. Models are fine-tuned on {10%, 30% and 60%} of the downstream dataset. Forecasting tasks involve varying forecasting lengths of {4, 96, 288, 672} time points and imputation tasks involve varying mask ratio {0.125, 0.25, 0.375, 0.5}. The length of the input window is 672. We average the result for each task.

| Model | Tasks | 60% | 30% | Decrease | 10% | Decrease |
|---|---|---|---|---|---|---|
| TS2vec | Forecasting(MSE) | 0.4723 | 0.5553 | 17.58% | 0.6275 | 32.87% |
| | Imputation(MSE) | 0.4021 | 0.4884 | 21.46% | 0.5739 | 42.72% |
| | Anomaly Detection(F0.5) | 0.4027 | 0.3454 | 14.24% | 0.3173 | 21.20% |
| | Classification(Acc.) | 0.5234 | 0.4197 | 19.82% | 0.4335 | 17.17% |
| CoST | Forecasting(MSE) | 0.4711 | 0.5589 | 18.64% | 0.6349 | 34.78% |
| | Imputation(MSE) | 0.3825 | 0.4704 | 22.97% | 0.5059 | 32.26% |
| | Anomaly Detection(F0.5) | 0.4221 | 0.3785 | *10.34% | 0.3156 | 25.23% |
| | Classification(Acc.) | 0.5534 | 0.4806 | 13.15% | 0.4363 | 21.15% |
| PatchTST | Forecasting(MSE) | 0.4456 | 0.5105 | 14.56% | 0.5716 | 28.29% |
| | Imputation(MSE) | 0.3623 | 0.4346 | 19.95% | 0.4592 | 26.76% |
| | Anomaly Detection(F0.5) | 0.3452 | 0.2657 | 23.03% | 0.2283 | 33.87% |
| | Classification(Acc.) | 0.4526 | 0.3341 | 26.18% | 0.2808 | 37.95% |
| UniTime | Forecasting(MSE) | 0.3904 | *0.4220 | 8.10% | 0.4528 | 15.98% |
| | Imputation(MSE) | 0.3375 | 0.3722 | 10.29% | 0.3895 | 15.41% |
| | Anomaly Detection(F0.5) | 0.4102 | 0.3640 | 11.26% | 0.3391 | 17.34% |
| | Classification(Acc.) | 0.5439 | 0.4740 | 12.85% | 0.4551 | 16.33% |
| TimeLLM | Forecasting(MSE) | 0.3713 | 0.4034 | *8.64% | 0.4180 | 12.58% |
| | Imputation(MSE) | 0.2815 | 0.3072 | 9.13% | 0.3104 | 10.27% |
| | Anomaly Detection(F0.5) | 0.4024 | 0.3655 | 9.16% | *0.3534 | 12.17% |
| | Classification(Acc.) | 0.5417 | 0.4958 | 8.48% | *0.4637 | *14.39% |
| GPT4TS | Forecasting(MSE) | *0.3838 | 0.4343 | 13.15% | *0.4447 | *15.86% |
| | Imputation(MSE) | *0.3212 | *0.3614 | 12.53% | *0.3846 | 19.75% |
| | Anomaly Detection(F0.5) | *0.4196 | *0.3718 | 11.39% | 0.3587 | *14.52% |
| | Classification(Acc.) | *0.5483 | *0.4902 | *10.60% | 0.4737 | 13.61% |
| PowerPM | Forecasting(MSE) | **0.3343** | **0.3551** | **6.22%** | **0.3652** | **9.25%** |
| | Imputation(MSE) | **0.2717** | **0.2998** | *10.34% | 0.3167 | *16.57% |
| | Anomaly Detection(F0.5) | **0.4822** | **0.4459** | **7.53%** | 0.4166 | 13.60% |
| | Classification(Acc.) | **0.6594** | **0.5943** | 9.88% | **0.5735** | **13.02%** |

