# OpenReview forum: "PowerPM: Foundation Model for Power Systems"
_NeurIPS.cc/2024/Conference — NeurIPS 2024 poster_

### Official Review · Reviewer_Vuy7 · 2024-07-11

**Soundness:** 3
**Presentation:** 3
**Contribution:** 3
**Rating:** 6
**Confidence:** 3

**Summary:**

The paper introduces the PowerPM: Foundation Model for Power Systems, which is designed to address the challenges of learning a generic representation of electricity time series (ETS) data in power systems. The model incorporates a temporal encoder and a hierarchical encoder to effectively capture the complex hierarchical structure and temporal dependencies present in ETS data. Additionally, PowerPM utilizes a self-supervised pre-training framework that includes masked ETS modeling and dual-view contrastive learning to enhance its ability to capture temporal dependencies and discrepancies across ETS windows. Overall, the contributions of the paper lie in the development of a sophisticated model that can accurately represent and analyze ETS data in power systems, offering valuable insights for various applications in the field.

**Strengths:**

### Strengths Assessment:

1. **Originality:**
   - The paper demonstrates originality in its approach to addressing the challenges of modeling electricity time series data in power systems. The combination of a temporal encoder and hierarchical encoder, along with the self-supervised pre-training framework, showcases innovative thinking in capturing temporal dependencies and hierarchical correlations simultaneously.
   - The use of masked ETS modeling and dual-view contrastive learning in the pre-training stage adds a novel dimension to the model's ability to learn universal representations from ETS data.

2. **Quality:**
   - The quality of the paper is evident in the thorough description of the PowerPM model, its components, and the experimental results obtained. The model's deployment in real-world scenarios and the achievement of state-of-the-art performance on diverse downstream tasks within private and public datasets reflect the high quality of the research.
   - The incorporation of extensive experiments, ablation studies, and few-shot experiments provides a robust evaluation of the model's effectiveness and generalization ability across various tasks and domains.

3. **Clarity:**
   - The paper is well-structured and clearly articulates the motivation, methodology, and results of the PowerPM model. The descriptions of the temporal encoder, hierarchical encoder, and the self-supervised pre-training framework are presented in a coherent manner, making it easy for readers to understand the technical aspects of the model.
   - The inclusion of figures, tables, and detailed explanations aids in clarifying complex concepts such as the model analysis, ablation study, and model scale evaluation.

4. **Significance:**
   - The significance of the paper lies in its contribution to advancing the modeling of electricity time series data in power systems. By introducing the PowerPM model, which effectively captures temporal dependencies and hierarchical correlations, the research offers a valuable tool for enhancing economic efficiency and promoting low-carbon principles in power systems.
   - The model's superior performance on diverse downstream tasks, its generalization ability across different datasets, and the economic benefits generated in real-world deployments underscore the practical significance of the research in the field of power systems modeling.

Overall, the paper excels in originality, quality, clarity, and significance, making a substantial contribution to the domain of electricity time series data modeling in power systems.

**Weaknesses:**

### Weaknesses Assessment:

1. **Limited Comparison with State-of-the-Art Models:**
   - While the paper highlights the superior performance of PowerPM over baseline models, there is a lack of comparison with the most recent state-of-the-art models in the field of electricity time series data analysis. Including comparisons with cutting-edge models could provide a more comprehensive evaluation of PowerPM's performance [T1].

2. **Insufficient Discussion on Model Interpretability:**
   - The paper could benefit from a more in-depth discussion on the interpretability of the PowerPM model. Providing insights into how the model captures and represents temporal dependencies and hierarchical correlations in ETS data could enhance the understanding of its inner workings and decision-making processes.

3. **Limited Exploration of Hyperparameters and Sensitivity Analysis:**
   - The paper lacks a detailed exploration of hyperparameters and sensitivity analysis for the PowerPM model. Including a thorough investigation of the impact of hyperparameters on model performance and conducting sensitivity analysis could provide valuable insights into the robustness and stability of the model across different settings.

4. **Scalability and Efficiency Considerations:**
   - The paper could further address scalability and efficiency considerations of the PowerPM model, especially when applied to large-scale datasets or real-time applications. Discussing the computational requirements, training time, and potential bottlenecks in scaling the model could help in understanding its practical feasibility in industrial settings.

5. **Limited Discussion on Ethical and Societal Implications:**
   - The paper could expand its discussion to include ethical and societal implications of deploying the PowerPM model in real-world power systems. Addressing issues related to data privacy, fairness, and potential biases in the model's predictions could enhance the overall impact and relevance of the research in a broader context.

### Suggestions for Improvement:

1. **Incorporate Comparison with State-of-the-Art Models:**
   - Conduct a thorough comparison with the latest state-of-the-art models in electricity time series analysis to provide a more comprehensive evaluation of PowerPM's performance and highlight its competitive advantages.

2. **Enhance Model Interpretability Analysis:**
   - Include a section dedicated to explaining how the PowerPM model interprets and processes ETS data, shedding light on its decision-making processes and enhancing the transparency of the model.

3. **Conduct Hyperparameter Tuning and Sensitivity Analysis:**
   - Perform a detailed hyperparameter tuning process and sensitivity analysis to understand the impact of key parameters on model performance and ensure robustness across different scenarios.

4. **Address Scalability and Efficiency Concerns:**
   - Discuss the scalability and efficiency aspects of the PowerPM model, including computational requirements, training time optimization strategies, and considerations for real-time deployment in power systems.

5. **Expand Discussion on Ethical and Societal Implications:**
   - Include a section on the ethical and societal implications of deploying the PowerPM model, addressing issues of fairness, bias, and privacy to ensure responsible and ethical use of the model in practical applications.

**Questions:**

### Questions and Suggestions for the Authors:

1. **Clarification on Model Interpretability:**
   - Can the authors provide more insights into how the PowerPM model interprets and captures temporal dependencies and hierarchical correlations in electricity time series data? Understanding the interpretability of the model can enhance its transparency and trustworthiness in real-world applications.

2. **Explanation on Hyperparameter Selection:**
   - Could the authors elaborate on the rationale behind the selection of specific hyperparameters for the PowerPM model? Providing details on the hyperparameter tuning process and its impact on model performance would offer valuable insights into the model's robustness.

3. **Discussion on Generalization to Unseen Domains:**
   - How does the PowerPM model generalize to unseen domains or datasets outside the ones used in the experiments? Understanding the model's ability to adapt to new data distributions and scenarios is crucial for assessing its practical utility in diverse real-world applications.

4. **Scalability and Deployment Considerations:**
   - What are the scalability considerations for deploying the PowerPM model in large-scale power systems? How does the model handle real-time data processing and what are the potential challenges in scaling it for industrial applications?

5. **Ethical and Fairness Implications:**
   - Have the authors considered the ethical implications of using the PowerPM model in power systems, particularly in terms of data privacy, fairness, and potential biases? Addressing these concerns can ensure responsible and unbiased deployment of the model in practice.

6. **Future Directions and Extensions:**
   - Are there any plans to extend the PowerPM model or explore new research directions based on the current findings? Discussing potential future developments or applications of the model can provide insights into the ongoing research trajectory in the field of power systems modeling.

7. **Comparison with Latest Research:**
   - Have the authors considered comparing the PowerPM model with the most recent state-of-the-art models in electricity time series analysis? Including such comparisons can strengthen the paper's contribution and highlight the advancements made by PowerPM in the field.

Addressing these questions and suggestions can enhance the clarity, robustness, and applicability of the PowerPM model, providing valuable insights for both the research community and practical stakeholders in the power systems domain.

**Limitations:**

### Assessment of Limitations and Societal Impact Considerations:

1. **Limitations Addressed:**
   - The authors have discussed the limitations of their work in Appendix F, which is a positive step towards transparency and acknowledging the constraints of the research. This demonstrates a willingness to reflect on the scope and potential weaknesses of the study.

2. **Societal Impact Considerations:**
   - The paper lacks a comprehensive discussion on the potential negative societal impacts of deploying the PowerPM model in real-world power systems. While some ethical considerations are mentioned, a more thorough exploration of broader societal implications is needed.

### Suggestions for Improvement:

1. **Enhanced Limitations Section:**
   - The authors could consider expanding the limitations section in the main body of the paper rather than relegating it to an appendix. This would ensure that readers engage with the limitations more prominently and understand the boundaries of the research.

2. **Broader Societal Impact Analysis:**
   - To address potential negative societal impacts, the authors should conduct a detailed analysis of how the deployment of the PowerPM model could affect various stakeholders, including issues related to fairness, privacy, and bias. Providing mitigation strategies for these impacts would demonstrate a proactive approach to responsible research.

3. **Incorporate Ethical Frameworks:**
   - Utilize established ethical frameworks or guidelines to evaluate the ethical implications of the PowerPM model in power systems. This could involve considering principles such as fairness, accountability, transparency, and privacy to ensure ethical deployment and minimize negative societal consequences.

4. **Engage with Stakeholders:**
   - Engage with relevant stakeholders, such as power system operators, regulators, and community representatives, to gather diverse perspectives on the potential societal impacts of implementing the PowerPM model. Incorporating feedback from stakeholders can enrich the discussion on societal implications.

5. **Mitigation Strategies:**
   - Propose concrete mitigation strategies for addressing any identified negative societal impacts of the PowerPM model. This could include mechanisms for monitoring model performance, ensuring fairness in decision-making processes, and enhancing transparency in model deployment.

By addressing these suggestions, the authors can strengthen the ethical foundation of their research, demonstrate a commitment to responsible innovation, and contribute to a more comprehensive understanding of the societal implications of deploying advanced models like PowerPM in practical settings.

---

> ### Author Rebuttal · Authors · 2024-08-07
>
> Thank you for your constructive and detailed comments. Responses to specific comments are listed below.
>
> **1. W1&Q7: Compare their framework with some state-of-art techniques**
>
> Thank you for your advice. Since ETS is essentially a time series, we choose the time series modeling model as the baseline, and all of its models are tested in the field of power systems in the original article.
>
> Our baseline includes not only the SOTA model of supervised learning, but also the SOTA model of self-supervised learning. With the application of large language models in time series, we also select three LLM enhanced Time series models of SOTA. See section "Experiment 3.1" and Appendix 4.1 for a description of baselines.
>
> Therefore, our experiment included three categories of SOTA models, and conducted sufficiently complete experiments, as shown in Table 1.
>
> **2. W2&Q1: Discussion on model interpretability**
>
> Thank you for your insightful comment on the interpretability of the PowerPM model. We acknowledge the importance of model interpretability and its impact on user trust and model adoption. Your feedback is valuable in guiding us to enhance the discussion on this aspect.
>
> In terms of model architecture, PowerPM consists of a temporal encoder and a hierarchical encoder, whose design goals are clear and public. Time encoders utilize Transformer encoders to capture temporal dependencies in time series and learn to predict future time points through mask ETS modeling strategies in the self-supervised learning framework.
>
> Hierarchical encoders use graph neural network technology, such as R-GCN, to simulate complex relationships between different hierarchy. This representation of hierarchical correlations is essential for understanding power system dynamics from both macro and micro perspectives.
>
> Our self-supervised learning framework further enhances the model's understanding of temporal dependence and differences across ETS windows. This not only improves the performance of the model, but also gives us a way to explain how the model learns these dependencies.
>
> **3. W3&Q2: Explanation on Hyperparameter Selection**
>
> Thank you for pointing out our problem. Due to the large number of parameters and huge pre-training data, and the long pretrain time of the model, considering that the existing foundation model work has not carried out the super-parameter search, for the sake of efficiency, the super-parameter in the pre-training stage is set by default with reference to the existing work.
>
> **3. W4&Q4: Scalability and Deployment Considerations**
>
> Thank you for raising these doubts. We have successfully deployed a lightweight version of PowerPM on the national grid for load forecasting.
>
> The main challenge encountered during the deployment process is that the inference resources of the National Grid are limited, resulting in low efficiency of the model operation and can not get real-time results. The solution is as follows:
>
> During the deployment phase, we trained models with different parameters for different tasks, including 120M and 250M, to achieve a balance of efficiency and accuracy. It is deployed on the grid's six NVIDIA 3090 for inference. For functions that are called more frequently, such as user load forecasting, we take into account efficiency factors and use a 120M model for inference. For functions that are called less frequently, such as area load forecasting, a 250M model can be used for inference, thus guaranteeing accuracy.
>
> At the same time, this scalability is very free, and the architecture of the model can be modified according to the number of parameters you want to adjust.
>
> **4. W5&Q5&L2: Ethics statement and social impact**
>
> The data collection and experiments conducted in our work have been approved by the Institutional Review Board (IRB) and passed ethical review. The data has been effectively licensed by State Grid Corporation of China in Zhejiang province, all user-related information is encrypted, and all downstream tasks are only used for State Grid supply planning and internal analysis.
> The social impact is shown as Appendix G.
>
> **5. Q3: Discussion on Generalization to Unseen Domains**
>
> Thank you for asking this question. Since PowerPM is a foundation model proposed for the power system, it cannot be generalized to data sets in the unknown field used in the experiment, but it can be generalized to any data set in the power system, which has been verified in the paper. We conducted training on private data sets and generalized it to four public datasets. The result is shown in Table 2.
>
> If the data set has a hierarchical relationship, the graph structure can be constructed. If there is no hierarchy (that is, the graph structure is an isolated graph), PowerPM can model normally without any modification. Therefore, PowerPM has a strong ability to adapt to new data distributions and scenarios in power system.
>
> **6. Q6: Future Directions and Extensions**
>
> Thank you for your interest in the future direction of our work. The future development of PowerPM is divided into the following two stages:
>
> $\bullet$ Stage 1: Deploy as many downstream tasks as possible to the National Grid for landing. This will make grid data analysis more intelligent and automated, and assist grid workers in decision-making, reducing costs and increasing efficiency.\
> $\bullet$ Stage 2: Currently, we only consider the macro-level interaction modeling. In the future, we will consider more factors, such as power grid topology, Kirchhoff's law and other constraints. To enhance the modeling process of ETS data to achieve a more general representation. This enhances downstream task performance and provides a more unique perspective for modeling ETS.
>
> Above is the future development direction of our work, thank you for your attention and support to our work.
>
> If you have any other concerns, please let us know what we can do to address any concerns you may have. Thanks for your thoughtful feedback again!

---

> > ### Comment · Area_Chair_Fzkc · 2024-08-12
> >
> > Dear Reviewer,
> >
> > As the discussion period is nearing its conclusion, we kindly ask you to engage in the discussion and provide notes on any concerns that have not yet been addressed, along with the reasons why.
> >
> > Thank you for your attention to this matter.
> >
> > AC.

---

> > > ### Comment · Area_Chair_Fzkc · 2024-08-13
> > > **Urgent: please respond to rebuttal**
> > >
> > > Dear Reviewer,
> > >
> > > As the discussion period is nearing its conclusion, we kindly ask you to engage in the discussion and provide notes on any concerns that have not yet been addressed, along with the reasons why.
> > >
> > > AC.

---

> ### Comment · Area_Chair_Fzkc · 2024-08-10
> **Please reply to the rebuttal.**
>
> Dear Reviewer,
>
> Please reply to the rebuttal.
>
> AC.

---

### Official Review · Reviewer_W2Rh · 2024-07-13

**Soundness:** 3
**Presentation:** 2
**Contribution:** 2
**Rating:** 3
**Confidence:** 4

**Summary:**

A pre-trained LLM named PowerPM is proposed for modeling Electricity Time Series (ETS) data. PowerPM combines a temporal encoder for capturing temporal patterns and a hierarchical encoder for understanding hierarchical correlations. PowerPM also employs a self-supervised pre-training strategy that incorporates masked ETS modeling and dual-view contrastive learning, enhancing the model's ability to learn from the intricacies of ETS data.

PowerPM is specifically designed for power systems, especially for the demand side, making it unique among other general models. This paper claims that deployment of PowerPM in Zhejiang Power Grid had led to significant economic benefits, showing the practical value of the model in real-world settings.

**Strengths:**

This article proposes a model for load time series forecasting on multi-hierarchy demand side and pre-trains it on a large amount of real data. It shows good performance and strong generalizability in experiments. Furthermore, the model is applicable in the real world.
For multiple pre-training challenges on multi-hierarchy load time series data, this method integrates corresponding solutions together.

**Weaknesses:**

Since the pre-training dataset is comprehensive, the work lacks an analysis on the effectiveness provided by the dataset. In other words, readers may not understand if the dataset works a lot or the proposed method.
The writing and explanation in this paper need to be improved.
This paper involves extensive training, but details are largely missing from the current paper.

**Questions:**

The writing lacks sufficient detail and explanation, resulting in quite some questions.
For example, what are the computational resources used? The provided materials do not include codes related to the proposed pre-training method. No information about whether the pre-trained model or the pre-training dataset will be released to make the work transparent.

**Limitations:**

This works involves load forecasting for power system and individual users in real-world applications, which would affect power system operation, its safety, and societal impact as a whole. These should be discussed.

---

> ### Author Rebuttal · Authors · 2024-08-07
>
> Thank you for your constructive and detailed comments. Responses to specific comments are listed below.
>
> **1. W1: About source of the model performance**
>
> Thank you for pointing out that our description may confuse the reader. We have made a detailed reply in **Global Response** , please check and wish it can answer your confusion.
>
> **2. Q1: About computational resources used**
>
> We are sorry that we don't provide clear guidance in the manuscript so that you missed this part. The computational costs and training process can be found in Appendix D.1. All the experiments are repeated five times, conducted on a Linux system with 2 CPUs (AMD EPYC 9654 96-Core Processor) and 8 GPUs (NVIDIA Tesla A800 80G) for about 8 days.
>
> **3. Q2: Codes related to the proposed pre-training method**
>
> Sorry we didn't have a clear description of the submitted code, so that you missed this pre-training method.
>
> The complete code has been placed in the supplementary material and has been uploaded at the time of the first submission of the manuscript. The file structure is as follows: (Hidden files have been ignored)
>
> .\
> ├── configs\
> ├── environment.yaml\
> ├── logs\
> ├── Makefile\
> ├── pyproject.toml\
> ├── result\
> ├── scripts\
> │   └── pretrain\
> │       └── run.sh\
> ├── src\
> │   ├── models\
> │   │   ├── powergpt_module_pretrain.py\
> │   │   └── PowerGPT.py\
> │   └── powergpt_pretrain.py\
> └── tests
>
> The entrance is in the process of the pretraining is **./ scripts/pretrain/run.sh**,  the script will execute python **./src/powergpt_pretrain. py**. Meanwhile, the **./src/models/powergpt_module_pretrain.py** file contains the pre-training process, where **loss_cl** and **loss_mse** are the loss functions corresponding to the two pretraining tasks.
> And **./src/models/PowerGPT.py** is the code file of our model backbone, which contains all the implementation code for the model and the detailed procedure for the pre-training tasks. That's all the relevant code and files for our pre-training.
>
> **4. Q3: Our planned steps for the dataset and pre-trained model release**
>
> We apologize for the temporary unavailability of the dataset, because it involves highly sensitive user electricity usage data and personal privacy concerns. But we will release our pre-trained model.
>
> In the future, in order to make this dataset a scientific research tool and better serve the research community, we plan to promote the release of our dataset in the following three stages:
>
> $\bullet$ Stage1: We plan to publicly release the pre-trained models with different scale in 250M、128M、64M and 35M, shortly after our work is accepted. And we also release the four public datasets. This will enable other researchers to not only replicate our model’s experimental results, but also utilize these models for other research of interest.
>
> $\bullet$ Stage2: We will actively communicate with the State Grid Corporation of China in Zhejiang province and aim to release the raw data of a portion of instances by the end of the year, to support further research efforts. The same as the work in our manuscript, these releases will be also conducted in compliance with ethical review requirements.
>
> $\bullet$ Stage3: In the future, we will explore the possibility of releasing the full dataset following approval of the relevant ethical review, to allow researchers to use the large-scale dataset for more research.
>
> **5. L1: Ethics statement**
>
> The data collection and experiments conducted in our work have been approved by the Institutional Review Board (IRB) and passed ethical review. The data has been effectively licensed by State Grid Corporation of China in Zhejiang province, all user-related information is encrypted, and all downstream tasks are only used for State Grid supply planning and internal analysis.
>
>
> If you still have any other new concerns, we would be eager to know what we can do to address any questions or concerns you may have. Thanks for your thoughtful feedback again!

---

> > ### Comment · Area_Chair_Fzkc · 2024-08-12
> >
> > Dear Reviewer,
> >
> > As the discussion period is nearing its conclusion, we kindly ask you to engage in the discussion and provide notes on any concerns that have not yet been addressed, along with the reasons why.
> >
> > Thank you for your attention to this matter.
> >
> > AC.

---

> > > ### Comment · Area_Chair_Fzkc · 2024-08-13
> > > **Urgent: please respond to rebuttal**
> > >
> > > Dear Reviewer,
> > >
> > > As the discussion period is nearing its conclusion, we kindly ask you to engage in the discussion and provide notes on any concerns that have not yet been addressed, along with the reasons why.
> > >
> > > AC.

---

> ### Comment · Area_Chair_Fzkc · 2024-08-10
> **Please reply to the rebuttal.**
>
> Dear Reviewer,
>
> Please reply to the rebuttal.
>
> AC.

---

### Official Review · Reviewer_Q8uG · 2024-07-13

**Soundness:** 3
**Presentation:** 2
**Contribution:** 3
**Rating:** 7
**Confidence:** 4

**Summary:**

This paper proposed a foundation model, PowerPM, to model electricity time series data, providing a large-scale off-the-shelf model for power systems. PowerPM consists of a temporal encoder and a hierarchical encoder with a self-supervised pretraining framework. The authors have tested PowerPM in Demand-side Management, Grid Stability, and Consumer Behavior Analysis, showing their advantages against other common time series LLM models, such as GPT4, TimeLLM, etc.

**Strengths:**

1. They have tested their work with real data and showing competitive results;
2. It is the first model that considers temporal dependency and hierarchical dependency simultaneously;
3. Comprehensive model analyses.

**Weaknesses:**

The writing should be improved, several places are missing space between words.
It could be improved or make the paper more convincing if the author could compare their framework with some state-of-art techniques. However, it is understandable that the research question/application objectives are more comprehensive in this paper.

**Questions:**

1. For power systems, have you considered the power systems constraints, such as power network structures, physic laws of Kirchhoff's circuit laws?
2. With renewable energy sources, have you considered of including weather time series data along with ETS data?

**Limitations:**

The authors has discussed the limitations of their work.

---

> ### Author Rebuttal · Authors · 2024-08-07
>
> We thank the reviewer for all the insightful comments. We apologize for the imprecise claims in certain parts of our manuscript and misunderstandings caused by our writing. Responses to specific comments are listed below.
>
> **1. W1: Several places are missing space between words**
>
> Thank you for pointing out our writing problems. We are sorry that we have caused you discomfort in reading due to our negligence. We've gone over The spelling issues and missing space, The updates are as follows:
>
> $\bullet$ in line 77 of the manuscript, "Model **(PowerPM),**" will be changed into "Model **(PowerPM) ,**"\
> $\bullet$ in line 77 of the manuscript, ”about **250Mparameters**“ will be changed into ”about **250M parameters**“.\
> $\bullet$ in line 763 of the manuscript, "pre-train on **Load** data." will be changed into " pre-train on **load** data."\
> $\bullet$ in the caption of figure 1, "(d)  Various tasks in power **systems**" will be changed into "(d)  Various tasks in power **systems.**"
>
> **2. W2: Compare their framework with some state-of-art techniques**
>
> Thank you for your advice. Since ETS is essentially a time series, we choose the time series modeling model as the baseline, and all of its models are tested in the field of power systems in the original article.
>
> Our baseline includes not only the SOTA model of supervised learning, but also the SOTA model of self-supervised learning. With the application of large language models in time series, we also select three LLM enhanced Time series models of SOTA. See section "Experiment 3.1" and Appendix 4.1 for a description of baselines.
>
> Therefore, our experiment included three categories of SOTA models, and conducted sufficiently complete experiments, as shown in Table 1.
>
> **3. Q1: Whether considered the power systems constraints**
>
> Thank you for your suggestion, we are sorry that we did not take into account the  power systems constraints, such as Kirchhoff's circuit laws, etc.
>
> Because all the data we currently obtain is provided by State Grid Corporation of China in Zhejiang province, it is the electricity consumption data after desensitization, which only contains the instance data recorded in the terminal meter and sensor. Its minimum granularity only sinks to the user unit and does not include complex circuit topological relationships within or between users. Therefore, we do not consider the constraints of the power network topology here, and only model ETS according to the physical level constraints in the real world, which can achieve quite superior results.
>
> But it's worth noting, your suggestion is very promising and meaningful, because considering more constraints can better model the power system. In the future, we will consider including power network structures, such asKirchhoff's circuit laws and other factors to model ETS. Please keep paying attention to our work and thank you again for your suggestions.
>
> **4. Q2: Whether considered of including weather time series data**
>
> Yes, as you said, we have considered weather time series data. Please refer to lines 135-142 in the "Method" section for the detailed description and figure 2(b) for the legend.
>
> We crawled the local weather and temperature on the public website, called as exogenous variables, and assigned the learnable parameters according to its different values and mapped to the embedding table. Each ETS windows has a corresponding sequence of exogenous variables. When the temporal encoder models ETS, the corresponding exogenous variables will look up the corresponding representation in the embedding table index, and finally add and fuse to obtain the ETS representation enhanced by external variables.
>
> Thank you again for your thorough consideration. The ablation experiment results are shown in Table 7 in the Appendix. After adding the weather and other exogenous variables, the performance of the model will indeed be improved to a certain extent, especially the Solar generation forecasting which is greatly affected by environmental factors.

---

> > ### Comment · Reviewer_Q8uG · 2024-08-09
> >
> > Awesome. Looking forward to following your future work!

---

> > > ### Author Response · Authors · 2024-08-10
> > > **Thank you**
> > >
> > > We wholeheartedly appreciate and deeply cherish your efforts in helping us to strengthen the paper and your recognition of our work.

---

### Official Review · Reviewer_vBXw · 2024-07-13

**Soundness:** 2
**Presentation:** 2
**Contribution:** 1
**Rating:** 3
**Confidence:** 3

**Summary:**

This paper learns a generic representation of electricity time series data. The proposed PowerPM model is composed of a temporal encoder and a hierarchical encoder.

**Strengths:**

The results shown in the table exhibit good numerical results of the proposed model.

**Weaknesses:**

It is unclear where the performance gain comes from. And the proposed architecture is a combination of several previous techniques, so it is hard to identify the technical contributions.

Moreover, it is not well motivated on the usage of developed models for many power system tasks. As far as the reviewer is aware of, current statistical methods and machine learning-based methods can already give good load forecasting results. The real challenges come from noisy data inputs, incomplete number of features, or small-region level forecasting. Yet this paper only looks into a very general forecasting problem, and does not reveal the key challenges and special properties of energy forecasting.

The grid stability notion is quite misleading. In standard power system tasks, stability notion is with respect to either frequency or voltage, and is more related to the transient states of the systems. While in this paper, the stability is more like a phenomenon or behavior analysis.

In addition, the paper needs to discuss the computational costs related to both training and inference, as utility and grid operators normally are not equipped with enough computation capabilities to use large foundation models.

**Questions:**

Can the authors elaborate the settings of the freeze version?

What are the effects of contrastive learning?

The figure 4 looks like a software snapshot. Can the authors explain the major information conveyed in this figure?

**Limitations:**

The paper describes techniques limited to energy forecasting tasks, and it is not clear if the methods can be applied to other tasks.

---

> ### Author Rebuttal · Authors · 2024-08-07
>
> Thank you for your constructive and detailed comments. We apologize for the imprecise claims in certain parts of our manuscript. We will reconsider the claims to be more precise. We hope that the responses below could address your specific comments.
>
> **1. W1: Clarify where the performance of the model comes from and the technical contribution of the model**
>
> Thank you for your valuable comments. Please refer to  **Global Response** for answers to these questions.
>
> **2. W2&L1: Only looks into a very general forecasting problem and disscuss generalization**
>
> Most of methods forecast on a single or small number of instances and have poor generalization. Due to the large difference in the electricity consumption of different users, the effect decreases significantly when it transfers to other regions. Secondly, most studies do not consider the correlation between hierarchies in ETS modeling. Our experimental results show that PowerPM achieves SOTA on the four public datasets, demonstrating strong generalization of the forecasting task and modeling hierarchical relationships can improve the effect of forecasting tasks.
>
> At the same time, **our model is not limited to forecasting tasks**, but can also perform tasks such as missing value imputation, electricity theft detection and so on. Through large-scale pre-training, PowerPM can successfully generalize to 44 power system tasks, and has a strong generalization ability, which is fundamentally different from the current end-to-end developed models for many power system tasks
>
> **3. W3: The grid stability notion is misleading**
>
> As you said, stability notion is with respect to either frequency or voltage, and  transient states of the system. And these tasks are also summarized around these aspects:
>
> $\bullet$ Electricity imputation:The missing load value will lead to misleading scheduling decisions, which will lead to the instability of the system voltage and destroy the grid stability[R1]. \
> $\bullet$ Clock anomaly detection:Clock anomalies will lead to clock frequency instability, resulting in synchronization errors, affecting the coordinated operation of power grid equipment, which will lead to instantaneous state out of control, affecting the grid stability[R2].\
> $\bullet$ Electricity theft detection: The behavior of stealing electricity will lead to the voltage change of local power grid, affecting the normal consumption of other users, which will also affect the grid stability[R3].
>
> Specifically, these three tasks are all about the frequency or voltage and the itransient states of the system, so we summarize them as grid stability.
>
> **4. W4: Disscuss on computational costs**
>
> Thank you for your valuable comments. Please refer to  **Global Response** for answers to these questions.
>
> **5. Q1: The effects of contrastive learning**
>
> Contrastive learning is used to capture the differences between different ETS windows. The results of Ablation experiments can be referred to in section 3.4 "Ablation Study", which shows that contrastive learning can effectively improve the performance of downstream tasks related to classification and anomaly detection. the related literature points that contrastive learning has assumed the downstream applications to be classifications[R4].
>
> **6. Q2: Elaborate the settings of the freeze version**
>
> We are sorry that we don't provide clear guidance in the manuscript so that you missed this part. The details of frozen version of PowerPM can be find in Appendix D.1, "Partial Fine-tuning" section.
>
> In the P-FT (Partial Fine-tuning) setup, for different tasks, we also introduce different head $H$ on the top of pre-trained encoder $f(.)$.
> For forecasting and imputation tasks, we use a prediction $H_l$  head to map prediction points or reconstruction points from $\mathbf{z}_i$.
> And for anomaly detection and classfication tasks, a classifier $H_c$ on top of the pre-trained encoder $f(.)$.
> During the whole finetune process, we keep the parameters of $f(.)$ fixed. Only the head is fine-tuned in this setup.
>
> **7. Q3: Explain the major information conveyed in this Figure 4**
>
> As you can see from the figure, the yellow bar chart indicates the electricity loss from actual orderly reductions  in electricity use, whereas the blue bar chart represents the electricity loss from scheduled  orderly reductions in electricity use. As observed from the figure, the scheduled orderly reductions predicted by PowerPM approach the energy loss resulting from actual reductions, which is sufficient to provide guidance for the planning of power systems, demonstrating the effectiveness of PowerPM. A detailed description can be found in Appendix B.
>
> If you still have any other new concerns, we would be eager to know what we can do to address any questions or concerns you may have. Thanks for your thoughtful feedback again!
>
> Reference:
>
> [R1] Wang M C, Tsai C F, Lin W C. Towards missing electric power data imputation for energy management systems[J]. Expert Systems with Applications, 2021, 174: 114743.
>
> [R2] Zhang H, Wang Q, Li Y, et al. Clock Anomaly Detection Method of Power Quality Monitoring Device Based on Voltage Sag[C]//2021 IEEE 2nd China International Youth Conference on Electrical Engineering (CIYCEE). IEEE, 2021: 1-6.
>
> [R3] Depuru S S S R, Wang L, Devabhaktuni V. Electricity theft: Overview, issues, prevention and a smart meter based approach to control theft[J]. Energy policy, 2011, 39(2): 1007-1015.
>
> [R4] Liu X, Zhang F, Hou Z, et al. Self-supervised learning: Generative or contrastive[J]. IEEE transactions on knowledge and data engineering, 2021, 35(1): 857-876.

---

> > ### Comment · Area_Chair_Fzkc · 2024-08-12
> >
> > Dear Reviewer,
> >
> > As the discussion period is nearing its conclusion, we kindly ask you to engage in the discussion and provide notes on any concerns that have not yet been addressed, along with the reasons why.
> >
> > Thank you for your attention to this matter.
> >
> > AC.

---

> > > ### Comment · Area_Chair_Fzkc · 2024-08-13
> > > **Urgent: please respond to rebuttal**
> > >
> > > Dear Reviewer,
> > >
> > > As the discussion period is nearing its conclusion, we kindly ask you to engage in the discussion and provide notes on any concerns that have not yet been addressed, along with the reasons why.
> > >
> > > AC.

---

> ### Comment · Area_Chair_Fzkc · 2024-08-10
> **Please reply to the rebuttal.**
>
> Dear Reviewer,
>
> Please reply to the rebuttal.
>
> AC.

---

### Author Rebuttal · Authors · 2024-08-07

## **Global Response to AC and all reviewers** ##
## About source of the model performance  ##
Thanks to all reviewers for the careful reading and thoughtful feedback. Here we explain the effectiveness of the dataset and the source of the model performance, as a solution to similar concerns raised by some reviewers. These concerns include:

$\bullet$ it is unclear where the performance gain comes from.(vBXw) \
$\bullet$ the lack of an analysis on the dataset effectiveness. Not understand if the dataset works a lot or the proposed method.(W2Rh)

**1. the effectiveness of dataset**

We regret that the following fact is not clearly explained in the manuscript: the pretrain dataset used in PowerPM is provided by State Grid Corporation of China in Zhejiang province, whose **authenticity** and **effectiveness** can be guaranteed. For a detailed description, please see Appendix C.

**2. source of the model performance**

We construct hierarchies for the dataset and introduce external weather variables, this is a feature of the dataset that was not included in previous work and is beneficial for ETS modeling. However, it is important to note that: to ensure the fairness of our experiments, all pretraining based baselines are re-pretrained on our pretraining dataset, instead of utilizing the weights published (if any) by the original work. In other words, all the performance comparison are conducted on the same data and with the similar schemes. We make fair comparisons not only on private datasets, but also on public datasets. The experimental results are shown in the Table 1 and Table 2.

Therefore, the performance improvement is mainly due to our superior models, which include temporal encoder and hierarchical encoder, and well-designed pre-training tasks, which can better utilize the characteristics of ETS. The advantages of each module can be highlighted in the ablation experiment, as shown in " 3.4 Model Analysis" section.

Therefore, the performance gains are due to our better model. Datasets play only a small part.

## About technical contribution ##
Above, we explain the source of the model performance, and here we summary the technical contribution to enable the reviewer to have a clearer understanding, And response the concern : The technical contribution is unclear.(vBXw)

The novelty of the proposed method are listed below:

$\bullet$ Self-supervised pre-training tasks:

Our method contains two novel self-supervised pretraining tasks(masked ETS modeling and dual-view contrastive learning), considering the characterics of ETS. The motivation of our designed tasks is to keep the  temporal dependency within continuous ETS windows and preserve the unique patterns of across different instance, which is more consistent with the real electricity consumption pattern.

$\bullet$ Model design:

$\circ$ In temporal encoder, we first propose to integrate exogenous variables with embedding table for ETS modeling. Different from other multi-variable time series modeling methods[40, 49, 44], our exogenous variables series are not input as ETS features, but are mapped as trainable parameters, which improves the flexibility and accuracy of ETS modeling.

$\circ$ In hierarchical encoder, we first propose the use of the real physical hierarchy to enhance ETS modeling. Different from other time series modeling methods with strong hierarchical constraints[R1-R3], we fully consider the characteristics of ETS data, propose to build hierarchical graph using region relationships, and use GNN to model these hierarchical correlation for the first time.

Benefit from the above technical contribution, we can achieve better performance in this scenario.

## About computational costs ##
We will explain the computational costs during train, inference and deployment to response the concern of reviewer vBXw,  W2Rh and Vuy7:

We are sorry that we don't provide clear guidance in the manuscript so that you missed this part. The computational costs and training process can be found in Appendix D.1.
All the experiments are repeated five times, conducted on a Linux  system with 2 CPUs (AMD EPYC 9654 96-Core Processor) and 8 GPUs (NVIDIA Tesla A800 80G)  for about 8 days.

We use a single NVIDIA 3090 for inference. According to different tasks, the inference time is about 1h for users' weekly load forecasting and 2s for city load forecasting.

During the deployment phase, we trained models with different parameters for different tasks, including 120M and 250M, to achieve a balance of efficiency and accuracy. It is deployed on the grid's six NVIDIA 3090 for inference.
For functions that are called more frequently, such as user load forecasting, we take into account efficiency factors and use a 120M model for inference.
For functions that are called less frequently, such as area load forecasting, a 250M model can be used for inference, thus guaranteeing accuracy.

## About Specific Responses ##

We have individually addressed all of your comments below, specifically addressing each reviewer’s concerns in the corresponding responses. Please note that in our responses, references in the format "[R1]" indicate citations that are newly added in the rebuttal, while references in the format "[1]" are citations from the original manuscript.

We have dedicated significant effort to improving our manuscript, and we sincerely hope that our responses will be informative and valuable. We would love to receive your further feedback.

Reference：

[R1] Orcutt G H, Watts H W, Edwards J B. Data aggregation and information loss[J].  The American Economic Review, 1968, 58(4): 773–787.

[R2] Anderer M, Li F. Hierarchical forecasting with a top-down alignment of independent-level forecasts[J]. International Journal of Forecasting, 2022, 38(4): 1405-1414.

[R3] Pang Y, Yao B, Zhou X, et al. Hierarchical Electricity Time Series Forecasting for Integrating Consumption Patterns Analysis and Aggregation Consistency[C]//IJCAI. 2018: 3506-3512.

---

### Decision · Program_Chairs · 2024-09-25

**Decision:**

Accept (poster)

**Comment:**

The paper proposes a pre-trained model for electricity time series data, combining temporal and hierarchical encoders to capture complex dependencies in power systems. Initial reviews were mixed, acknowledging comprehensive experiments and potential practical impact, but raising concerns about the source of performance gains, computational costs, and comparisons with state-of-the-art methods.

In their rebuttal, the authors provided detailed responses, clarifying performance gains, computational costs, and ethical considerations. They also outlined plans for model and dataset release.

Two reviewers who recommended rejection did not respond to the rebuttal despite repeated reminders. Their concerns were addressed by the rebuttal. The paper also went through extensive ethical reviews. The concerns of the ethical reviewers were also addressed. Therefore I tend to side with the two positive reviewers and recommend to accept this submission due to its solid methodology and especially its potential practical impact.

I read the paper myself and am impressed by its practical real applications. Due to the fact that I overruled the two negative but non-responsive reviewers, I discussed with the senior AC about this paper. The SAC agreed with my decision.